# Geometric control of myosin II orientation during axis elongation

**Matthew F Lefebvre[1]\*[†], Nikolas H Claussen[1]\*[†], Noah P Mitchell[1,2], Hannah J Gustafson[1,3], Sebastian J Streichan[1]\***

[1]Department of Physics, University of California, Santa Barbara, Santa Barbara, United States; [2]Kavli Institute for Theoretical Physics, University of California, Santa Barbara, Santa Barbara, United States; [3]Biomolecular Science and Engineering, University of California, Santa Barbara, Santa Barbara, United States

**Abstract** The actomyosin cytoskeleton is a crucial driver of morphogenesis. Yet how the behavior of large-scale cytoskeletal patterns in deforming tissues emerges from the interplay of geometry, genetics, and mechanics remains incompletely understood. Convergent extension in *Drosophila melanogaster* embryos provides the opportunity to establish a quantitative understanding of the dynamics of anisotropic non-muscle myosin II. Cell-scale analysis of protein localization in fixed embryos suggests that gene expression patterns govern myosin anisotropy via complex rules. However, technical limitations have impeded quantitative and dynamic studies of this process at the whole embryo level, leaving the role of geometry open. Here, we combine in toto live imaging with quantitative analysis of molecular dynamics to characterize the distribution of myosin anisotropy and the corresponding genetic patterning. We found pair rule gene expression continuously deformed, flowing with the tissue frame. In contrast, myosin anisotropy orientation remained approximately static and was only weakly deflected from the stationary dorsal-ventral axis of the embryo. We propose that myosin is recruited by a geometrically defined static source, potentially related to the embryo-scale epithelial tension, and account for transient deflections by cytoskeletal turnover and junction reorientation by flow. With only one parameter, this model quantitatively accounts for the time course of myosin anisotropy orientation in wild-type, *twist*, and *even-skipped* embryos, as well as embryos with perturbed egg geometry. Geometric patterning of the cytoskeleton suggests a simple physical strategy to ensure a robust flow and formation of shape.

**\*For correspondence:**
mfl2@ucsb.edu (MFL);
nclaussen@ucsb.edu (NHC);
streicha@ucsb.edu (SJS)

[†]These authors contributed equally to this work

**Competing interest:** The authors declare that no competing interests exist.

## Editor's evaluation

This article reports fundamental findings regarding spatiotemporal control of myosin-based force generation during *Drosophila* germband extension and is of considerable interest to our understanding of tissue morphogenesis during early development. Using quantitative imaging, mathematical modeling, and mutant analysis, the authors provide compelling evidence that myosin polarity patterns are not governed by pair-rule gene expression, but that a geometric cue promotes myosin II accumulation of vertically oriented junctions. The results challenge current views of how gene expression patterns control myosin II anisotropies and provide new testable hypotheses on the role and importance of tissue geometry.

## Introduction

During morphogenesis, tissues dynamically remodel through cellular flows (*Collinet and Lecuit, 2021*). These flows are driven by patterned cytoskeletal processes, such as large-scale gradients of non-muscle myosin II (myosin), that generate imbalanced forces (*Heer and Martin, 2017*). Two

processes affect gene expression and cytoskeletal patterns during morphogenesis. First, cells move, taking their constituents with them. Second, the contents of cells are constantly reorganizing (*Garcia et al., 2013*; *Coravos et al., 2017*). If intracellular turnover is slow compared to the rate of tissue movement, pattern change is dominated by passive advection, and the pattern will remain stationary in the 'Lagrangian' (*Landau and Lifshitz, 1987*) frame of reference moving with the tissue.

Recent technological advances now allow the study of dynamic patterns at a global scale. In this way, it becomes possible to elucidate the effects of turnover and tissue-scale cues – relating dynamics of gene expression patterns, cytoskeletal components, and tissue shape during morphogenesis.

Here, we analyze the dynamics of myosin in *Drosophila melanogaster* to establish a quantitatively testable link between genetic patterning and organ geometry. We study the dynamics of the anisotropic distribution of myosin that drives global tissue flow through cell intercalation during body axis elongation (*Bertet et al., 2004*; *Blankenship et al., 2006*; *Rauzi et al., 2008*; *Paré and Zallen, 2020*), known as germband extension (GBE). Qualitative analysis of fixed embryos has been used to suggest a link between spatial patterns of pair rule genes (PRGs) and the orientation of myosin-rich junctions (MRJs), likely via members of the Toll-like transmembrane receptor (TLR) family (*Irvine and Wieschaus, 1994*; *Zallen and Wieschaus, 2004*; *Paré et al., 2014*; *Paré et al., 2019*; *Lavalou et al., 2021*). However, cell-scale live imaging demonstrates reorientation of myosin anisotropy by other factors, for example, mechanosensation (*Figure 1a*, *Fernandez-Gonzalez et al., 2009*; *Farrell et al., 2017*).

Using in toto light sheet microscopy and tissue cartography (*Krzic et al., 2012*; *Heemskerk and Streichan, 2015*), we map the time course of myosin, PRGs, and TLRs during GBE (*Figure 1c and d*). These maps show quantitatively that PRG and TLR expression patterns deform with tissue flow, whereas myosin orientation is only transiently deflected away from stationary geometric landmarks in response to flow. This leads to an increasing discrepancy between the pattern of PRG expression and myosin orientation over the course of GBE. We quantitatively explain the short-lived anisotropy deflection by the finite time of association (~5 min) between myosin motors and the junctional actin cortex. These results demonstrate that PRGs and Tartan, a receptor known to act in concert with TLRs, are passively advected by tissue flow, while the recruitment of myosin that drives flow is controlled by nearly static factors, in spite of significant tissue rearrangement.

## Results

### A quantitative mismatch between junctional myosin accumulation and PRG gradient patterns

Extensive analysis of fixed embryos indicates a link between PRG and TLR expression patterns, on the one hand, and anisotropic actomyosin organization, on the other hand (*Zallen and Wieschaus, 2004*; *Paré et al., 2014*; *Munjal et al., 2015*; *Paré et al., 2019*). The mechanism how TLR interaction recruits cytoskeletal components to adherens junctions however remains unclear (*Paré and Zallen, 2020*). Here, we extend this body of work by a quantitative, hypothesis-driven analysis of the dynamics of both myosin and PRGs at the whole embryo level during GBE. We digitally stitched data gathered from multiple live and fixed embryos to create a dynamic atlas comprising components of the anterior-posterior (AP) patterning system as well as myosin (*Figure 1c and d*), measured across the entire embryo (*Mitchell et al., 2022*). The time course of these gene products is provided with ~1 min temporal resolution, starting from cellularization until the end of GBE (see Appendix for detail). We define $t = 0$ to be the initiation of ventral furrow (VF) formation.

Throughout embryogenesis, egg geometry remains static (*Figure 1b*), defining a fixed reference frame that is described by a coordinate system parallel to the anterior-posterior (AP) and dorsal-ventral (DV) axes. All 3D renderings and whole-embryo cylindrical projections in this paper are oriented as in *Figure 1b*. We focus on junctional myosin at the apical surface (*Paré and Zallen, 2020*, *Figure 1c*), together with the PRGs Even-Skipped (Eve), Runt, Fushi-Tarazu (Ftz), Hairy, Paired (Prd), and Sloppy-Paired (Slp), to create a dynamic atlas of gene expression during GBE (*Figure 1d*). As reported previously, MRJs mainly align with the DV axis (*Figure 1c*, *Bertet et al., 2004*), while all of the PRGs we analyzed were expressed in a series of stripes occurring at regular intervals along the AP axis (*Figure 1d*, *Irvine and Wieschaus, 1994*; *Clark and Akam, 2016*). We mine this expression atlas

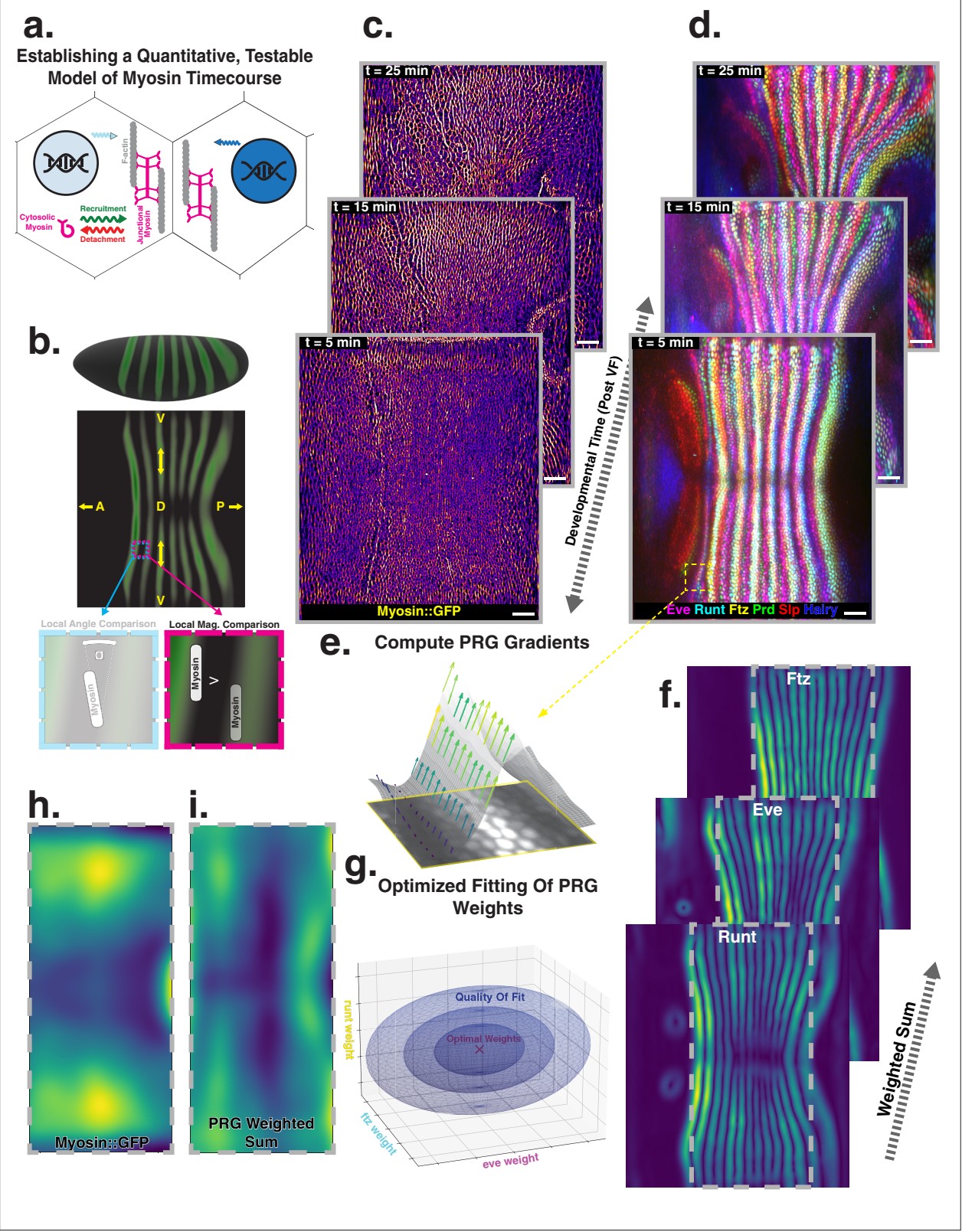

**Figure 1.** Global analysis of myosin vs. pair rule gene (PRG) expression patterns reveals no linear correlation. (**a**) Junctional myosin could be regulated by gene expression patterns, by mechanical cues, or both. (**b**) Tissue cartography extracts embryo surfaces from volumetric in toto light sheet imaging that are projected onto a cylindrical chart. This allows measuring quantities on a tissue scale, here the intensity and orientation of junctional myosin and PRGs. All figures follow the orientation indicated here: anterior left, posterior right. For 3D-rendered embryos, dorsal is up, and for full-embryo

*Figure 1 continued on next page*

*Figure 1 continued*

cylindrical projections, dorsal is in the middle. (**c**) Time series of junctional myosin, starting at 5 min post ventral furrow initiation (**d**) Time series of PRGs, starting at 5 min post ventral furrow initiation. Time series created by digitally stitching together different stained and live imaged embryos. (**e**) The smoothed gradient of PRG expression patterns computes local cell–cell differences. The gradient vector points in the direction in which the signal increases most. (**f**) Gradient magnitude of the expression patterns of the PRGs Runt, Eve, and Ftz in three representative embryos shows 14 stripes with a dorsal-ventral (DV) modulation of intensity. (**g**) In regression analysis, the smoothed gradients of PRGs are combined into a weighted sum to approximate the observed myosin pattern. The weights are adjusted to optimize the quality of fit, exploring the entire space of possible weights, both positive and negative. (**h**) Smoothed junctional myosin intensity at the onset of germband extension (GBE) (ensemble average of five embryos). (**i**) Result of PRG gradient regression. The best possible fit using the weighted sum of PRG gradients does not resemble the large-scale myosin pattern.

to quantitatively test the relationship between the accumulation of myosin on junctions and cumulative PRG expression (see 'Quantitative analysis of junctional myosin').

Differences of PRG expression levels in adjacent cells have been proposed to underlie anisotropic myosin accumulation (*Zallen and Wieschaus, 2004*; *Paré and Zallen, 2020*). This provides a testable prediction relating PRGs to the accumulated signal and orientation of MRJs. We first computed local differences (gradients) of PRG expression levels, focusing on Runt and Eve – which are known to have the strongest impact on GBE (*Irvine and Wieschaus, 1994*) – as well as Ftz. Gradients were steepest along the AP axis, pointing towards the center of the stripes (*Figure 1e*). Across the germband, the magnitude of differences showed 14 regularly spaced stripes. While all PRGs are slightly out of register with one another (*Clark and Akam, 2016*), the six PRGs we analyzed had the following characteristics: (i) along the AP axis, gradients were strongest at the first and last stripe (*Petkova et al., 2019*), and (ii) expression levels reduced towards the dorsal pole (*Figure 1f*).

The profile of junctional myosin also demonstrated a DV gradient similar to the one observed for the PRGs, with a minimum on the dorsal pole (*Streichan et al., 2018*). However, along the AP axis, the myosin profile did not match any individual PRG gradient because they all have gaps in intensity between stripes (*Figure 1c*), while myosin does not. Therefore, we investigated whether combining the gradient profiles of multiple PRGs could produce a pattern consistent with the observed myosin accumulation. We used linear regression to compare the observed magnitude of junctional myosin with a weighted sum of PRG gradient patterns. The weighting parameters are adjusted to achieve the best possible agreement with the myosin profile (*Equation 1*).

$$\text{myosin} = \sum_{\text{PRGs}} \text{PRG gradient} \times \text{weighting parameter} \tag{1}$$

Each PRG had its own parameter, which did not change over space or between different stripes. It was permitted to be positive or negative, representing a promotion or inhibition of myosin accumulation (for details, see 'PRG gradient regression'). The PRG regression can also be seen as an analysis of the large-scale correlation between junctional myosin and PRG gradients, without making any a priori assumptions about the individual effects of each PRG.

The best fit produced in this way captured the observed DV modulation of myosin (*Figure 1h and i*). Along the AP axis, however, the patterns were quantitatively and qualitatively different. The model of local PRG differences predicted strong myosin accumulations at the anterior and posterior ends of the germband and a minimum in the center (*Figure 1i*). This did not fit the observed myosin localization pattern (*Figure 1h*). *Equation 1* is the simplest possible way to model a potential relationship between myosin and PRG gradients. In the appendix ('Nonlinear PRG-based model'), we analyze more complex, nonlinear regression models, and show that they also fail to account for the observed myosin pattern.

Our analysis suggests that local differences of PRGs cannot be linearly related to the amount of myosin accumulating on junctions, without postulating currently unknown additional complexity.

## Reorientation of PRGs by advection vs. near-stationary orientation of myosin anisotropy

Anisotropic myosin distributions are characterized by both a local intensity – which did not correlate with PRG patterns – and a local orientation, to which we turn next. If myosin recruitment is instructed genetically according to time-independent rules, one would expect the angle between the myosin orientation and the gene stripes to be constant. Therefore, we analyzed the local orientation of MRJs

across the whole embryo surface over time, which we compared to the concurrent orientation of local PRG stripes.

During GBE, both MRJs and PRG orientations are under the influence of significant tissue flow. As a consequence, their patterns could deform with the flow (i.e., passive advection behavior). We characterized the time course of the instantaneous flow field, which quantifies the global pattern of cell movements. We compared this quantification of flow with the temporal evolution of the localization patterns of PRGs and TLRs (*Figure 2a–c'*).

Just after the initiation of gastrulation (7 min post VF initiation), Runt localization in the germband is characterized by seven stripes at different AP positions (*Figure 2a and b*, Appendix 1; *Clark and Akam, 2016*). Since this pattern of expression is stereotypic for the PRGs known to have the largest individual effects on GBE (*Clark and Akam, 2016*; *Irvine and Wieschaus, 1994*), we adopted Runt as our representative PRG. In *Appendix 1—figure 10* and *Figure 1d*, we show that during GBE, the stripes of the PRGS Runt, Eve, Ftz, Paired, Sloppy-Paired, and Hairy remain parallel throughout GBE flow, and are transported by tissue flow in the same way. Therefore, it is sufficient to study the advection behavior of only one of them. Similarly, as a proxy for the TLRs, we chose Tartan (*Paré et al., 2019*), a leucine-rich-repeat receptor downstream of the PRGs that has been implicated in directing myosin anisotropy in concert with the TLRs 2, 6, and 8, due to the availability of a high-quality antibody.

15 minutes later (22 min post VF initiation), the PRG and Tartan stripes were strongly deformed, with the posterior-most Runt stripe (stripe 7) almost completely moved onto the dorsal side of the embryo. We also tested whether the dynamic expression of Tartans matched what is observed for PRGs. We found that the orientation of Tartan stripes (*Paré et al., 2019*), closely mirrored that of the Runt stripes (*Figure 2a' and b'*, Appendix 1).

The reorientation of PRG and Tartan localization during GBE is reflected in a rapid decline of the autocorrelation of local stripe orientations (*Figure 2f*). To understand whether tissue flow could account for this deformation, we calculated six inter-stripe positions of the Runt pattern (*Figure 2a*, magenta lines, *Appendix 1—figure 4*), which we advected along cell trajectories computed from the velocity field, as measured by particle image velocimetry (PIV). The resulting advected inter-stripe positions (*Figure 2a'*, cyan lines) remained between the position of Runt stripes at 22 min. Manual tracking (*Appendix 1—figure 5*) confirms that cells that initially express Runt retain expression after 20 min. Together, this suggests that both PRG and Tartan patterns flow with the tissue frame of reference: their reorientation is quantitatively accounted for by advection due to tissue flow.

Next, we asked whether the velocity field behaved in the same manner. 7 min post VF initiation, a characteristic tissue flow pattern emerged, with four vortices and two hyperbolic fixed points (*Streichan et al., 2018*, *Figure 2c*). This characteristic flow pattern persisted during GBE, as highlighted by a very similar pattern observed 15 min later (*Figure 2c'*). While the magnitude of the flow clearly changed, its local direction was nearly constant, and vortices shifted only slightly over time. The autocorrelation of local tissue flow direction remained high throughout GBE (*Figure 2e*). Thus, the instantaneous flow field was nearly stationary for the entire duration of GBE: although cells travel long distances across the embryo surface, different cell neighborhoods that pass through a common spatial coordinate will move in the same direction, irrespective of the precise time point during GBE.

Tissue flow is known to be driven by anisotropic myosin (*Bertet et al., 2004*; *Blankenship et al., 2006*; *Rauzi et al., 2008*; *Paré and Zallen, 2020*). In fact, the flow field can be quantitatively predicted from the myosin distribution using a simple hydrodynamic model (*Streichan et al., 2018*). This suggests that a static myosin pattern should be required to produce the observed, nearly static, tissue flow field. Yet directional myosin recruitment by cell-intrinsic PRG patterning should lead to a continuous reorientation and advection of myosin anisotropy.

To resolve this discrepancy, we measured the orientation of MRJs and compared them to local Runt stripe orientations (*Figure 2h–k'*). The latter is defined by the direction of the Runt gradient, rotated by 90°. MRJs were detected using a segmentation-free method based on the Radon transform (*Figure 2j and l*, *Streichan et al., 2018*; see 'Quantitative analysis of junctional myosin'). This method is robust even at low signal-to-noise ratios. The local myosin anisotropy direction (*Figure 2k–k'*) was defined by the intensity-weighted mean of MRJ orientations in a three-cell radius (see 'Quantitative analysis of junctional myosin'). To link local angles with the organ-scale geometry, we measured angles with respect to the DV axis, which we defined geometrically by the direction of maximal curvature of the embryo surface.

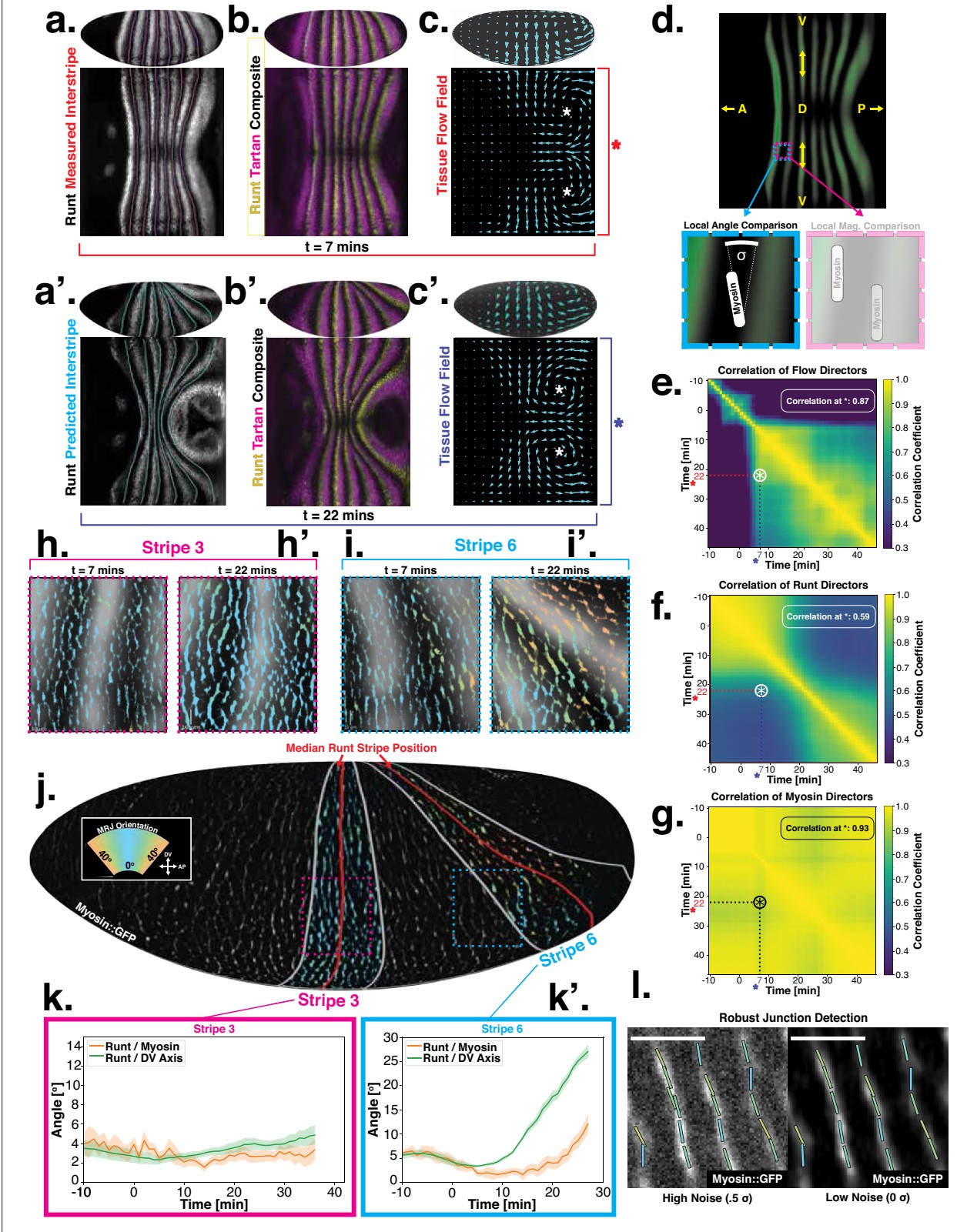

**Figure 2.** Pair rule genes (PRGs) flow with tissue while myosin pattern does not. (**a**) Runt stripes with measured inter-stripe lines, 7 min post ventral furrow (VF) initiation from a representative Runt::LlamaTag-GFP embryo. All PRG stripes are initially approximately parallel to the dorsal-ventral (DV) axis. (**a'**) PRG stripes deform due to advection by tissue flow. Runt stripes with inter-stripe lines predicted by advection, 22 min post VF initiation. Same embryo as in (**a**). (**b, b'**) Digitally stitched Runt/Tartan composite, 7 min (**b**) and 22 min (**b'**) post VF initiation, showing that PRG and TLR stripes remain

*Figure 2 continued on next page*

*Figure 2 continued*

parallel. (**c, c'**) Tissue flow field, 7 min (**c**) and 22 min (**c'**) post VF initiation. Calculated from an average of 5 WT Myosin::GFP embryos. (**d**) Using tissue cartography, we compare MRJ and PRG orientation across the entire embryo. (**e**) Temporal autocorrelation of the tissue flow field. Each pixel in the matrix shows the correlation (similarity in direction, ranging from 0 to 1, averaged over the embryo surface) of the flow fields at two different time points. (**f**) Temporal autocorrelation of the Runt stripe direction shows rapid decay during tissue flow. Data from 5 WT Runt::LlamaTag-GFP embryos (see Appendix 'Definition of Runt stripe angle' and 'Definition of correlation coefficient for nematic fields' for mathematical details). (**g**) Temporal autocorrelation of the tissue-scale myosin direction, showing an approximately static pattern of myosin orientation. Data from 5 WT Myosin::GFP embryos. (**h–h', i–i'**) Digitally stitched images showing Runt and junctional myosin (colored according to angle with DV axis) in a part of the regions surrounding runt stripes 3 (**h-h'**) and 6 (**i-i'**) at 7 min (**h,i**) and 22 min (**h', i'**) post VF initiation. In regions where gene patterns are deformed by flow, an angle discrepancy between myosin orientation and PRG stripes develops. Orientation is dorsal up, anterior left. (**j**) Junctional myosin at 22 min post VF initiation in a representative WT Myosin::GFP embryo. In highlighted regions (defined by Runt stripes 3 and 6), junction color corresponds to the junction/DV axis angle and junction brightness to the myosin fluorescent intensity. Red lines show median runt stripe position. (**k–k'**) Angle between myosin anisotropy orientation and Runt stripe, and angle between Runt stripe and DV axis, averaged over the regions corresponding to runt stripes 3 and 6. Runt angle measured by the direction of Runt gradient, rotated by 90°. (**l**) The radon transform method detects MRJs and is insensitive to noise. Bars indicate junctions detected by the radon transform, color-coded according to their angle. Orientation is dorsal up, anterior left. Scale bar $10\mu\text{m}$.

The degree to which a given PRG stripe deforms over time is dependent upon its position along the AP axis. In general, the closer a stripe is to the posterior pole of the embryo, the more it will deform due to the strong posterior vortices. Therefore, we analyzed the relationship between myosin and PRG orientations on a per-stripe basis (***Figure 2j***, SI), using Runt as a representative PRG.

For Runt stripe 3, MRJs are parallel to the DV axis (***Figure 2h, h' and k***), in accordance with previous reports (***Zallen and Wieschaus, 2004***; ***Paré et al., 2014***). As highlighted by Runt stripe 6, in more posterior stripes we observed an increasing myosin/DV axis angle (***Figure 2i, i' and k'***). In this region, 22 min into GBE, MRJ orientation was less streamlined than at the onset (***Figure 2i'***). Quantitatively, we found that the myosin/DV axis angle in Runt stripe 3 was small throughout GBE and aligned with MRJs (***Figure 2k***). In contrast, in Runt stripe 6, which rotated away from the DV axis due to its proximity to the posterior flow field vortex, MRJs reoriented away from the stripe (***Figure 2k'***). The global autocorrelation of myosin anisotropy orientation remained consistently high throughout GBE, indicating a nearly stationary pattern (***Figure 2g***), akin to instantaneous flow (***Figure 2e***). Finally, in ***Appendix 1—figure 15*** we show that in accordance with the above analyses, the rate of change of the Runt pattern is much lower in the Lagrangian frame of reference that flows with the tissue than in a static frame of reference, whereas the opposite is true for the myosin orientation.

How to establish an instructive link between a continuously reorienting PRG pattern that moves with the tissue frame of reference, and the nearly stationary direction of both the myosin anisotropy and the flow field is not clear. The dynamic, orientational mismatch is independent of what type of linear or non-linear form of PRG-instructed myosin recruitment is posited. Our observations raise the question: what intracellular dynamic rules for myosin recruitment are required to establish the observed global myosin pattern?

## Myosin dynamics are due to tissue-flow driven reorientation and subcellular turnover

To decode the nearly stationary myosin orientation and understand its residual dynamics, we studied the interplay of two dynamic effects: (i) at the subcellular level, the cytoskeleton can dynamically rearrange, due to binding and unbinding of molecular motors to the actin meshwork (***Agarwal and Zaidel-Bar, 2019***), and (ii) at the tissue level, advection will reorient junctions. We first studied myosin turnover using fluorescence recovery after photobleaching (FRAP). We photobleached individual junctions and measured the signal recovery for about 5 min (***Figure 3a***). In the first frame after bleaching (1.5 s), we measured approximately 50% signal reduction, indicating the presence of a fully mobile myosin subpopulation, possibly cytoplasmic (***Wachsmuth, 2014***). The recovery curve of $N = 25$ junctions reflected multiple time scales and had a high standard deviation, possibly due to myosin oscillations on a time scale of ~60 s (***Figure 3a'***, ***Gustafson et al., 2021***). During the first 30 s after photobleaching, fluorescence recovery was rapid, as previously measured (***Fernandez-Gonzalez et al., 2009***; ***Munjal et al., 2015***). After 30 s, the rate of recovery slowed, and pre-bleach levels were reached by 210 s, suggesting myosin molecular motors on junctions are dynamically recruited into the cortex and only bind transiently. See ***Appendix 1—figure 13*** for a a mathematical model of the FRAP recovery curve.

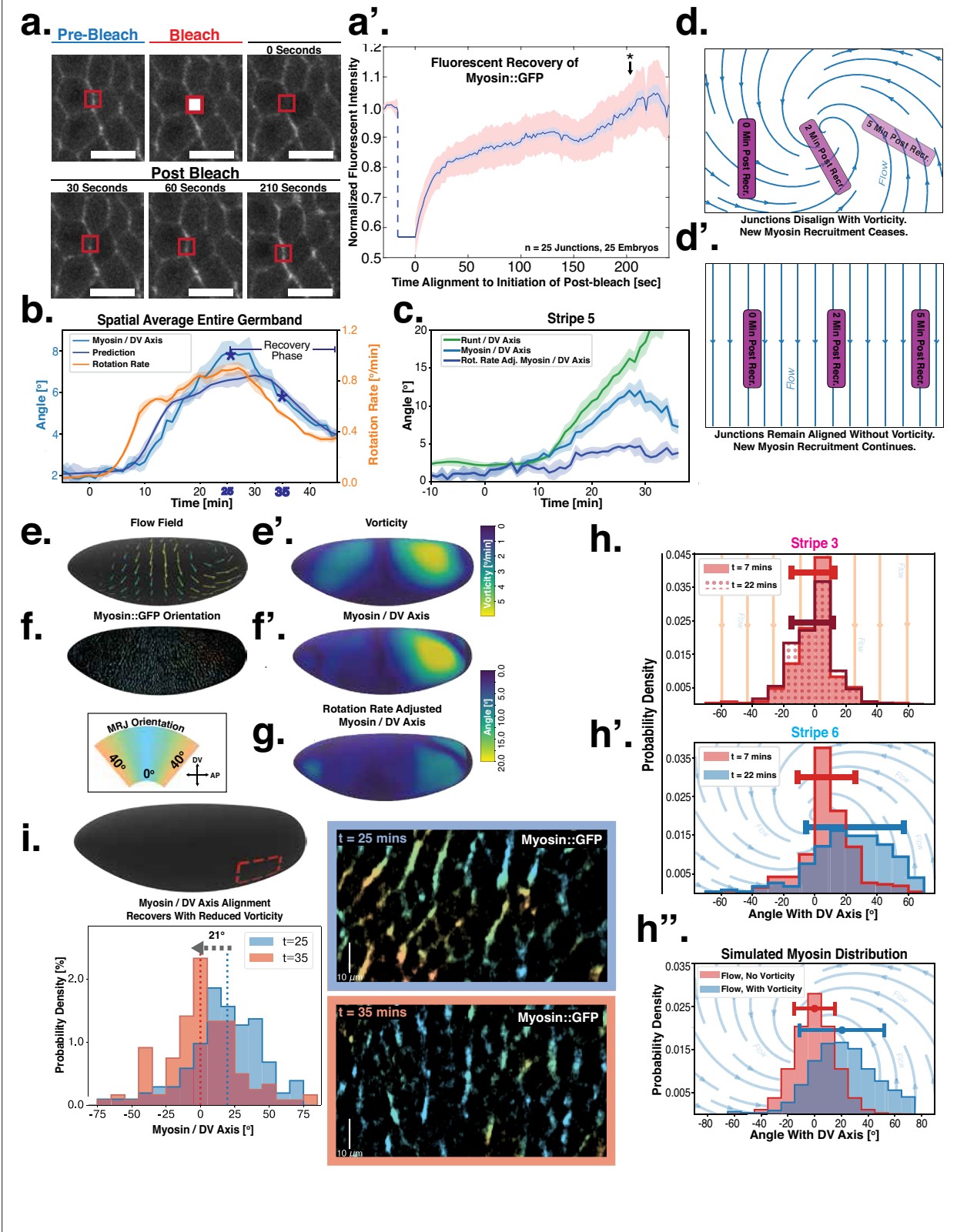

**Figure 3.** Dynamics of myosin orientation can be quantitatively captured by embryo geometry and vorticity. (**a**) Time series of a representative fluorescence recovery after photo bleaching (FRAP) experiment of junctional myosin. (**a'**) FRAP of junctional myosin ($N=25$ embryos) shows multiple timescales and complete recovery of myosin fluorescence, indicating transient binding to the cortex. Rose shaded area indicates standard deviation and blue shaded error the standard error of the mean. * indicates the time of full recovery. (**b**) Spatial average of vorticity and myosin/dorsal-ventral (DV)

*Figure 3 continued on next page*

*Figure 3 continued*

axis angle across germband versus time in $N = 5$ WT Myosin::GFP embryos. Blue line shows prediction for the myosin/DV axis angle calculated from vorticity. Stars highlight times shown in panel (**i**), showing myosin/DV axis alignment recovery. (**c**) Spatial average of Runt/DV axis angle, myosin/Runt stripe, myosin/DV axis angle, and rotation-rate corrected myosin/DV axis angle over the region corresponding to Runt stripe 5 versus time in $N = 5$ WT Myosin::GFP and $N = 5$ WT Runt::LlamaTag-GFP embryos. (**d, d'**) Myosin-rich junctions (MRJs) deflect away from the axis of preferential recruitment in rotational (**d**) flow but remain aligned in irrotational (**d'**) flow. (**e**) Tissue flow field during germband extension (GBE), temporal average from 15 to 25 min post VF initiation. Computed from ensemble of $N = 5$ WT Myosin:GFP embryos. (**e'**) Vorticity of tissue flow field, temporal average from 15 to 25 min post VF initiation. Computed from ensemble of $N = 5$ WT Myosin::GFP embryos. (**f**) MRJs in a representative WT Myosin::GFP embryo 22 min post VF initiation. Junction color corresponds to the junction/DV axis angle and junction brightness to the myosin fluorescent intensity. (**f'**) Smoothed myosin/DV axis angle, temporal average from 15 to 25 min post VF initiation. Computed from ensemble of $N = 5$ WT Myosin::GFP embryos. (**g**) Rotation rate-adjusted myosin/DV axis angle, temporal average from 15 to 25 min post VF initiation. Computed from ensemble of $N = 5$ WT Myosin::GFP embryos. (**h, h'**) Histogram of myosin angular distribution in the region corresponding to Runt stripe 3 (**h**) and stripe 6 (**h'**), at two times during GBE. Data corresponds to region shown in *Figure 2h'–i'* (one representative WT Myosin::GFP embryo). Histograms colored in shades of red, resp. blue, show data at times and regions where vorticity is low, resp. high. (**h''**) Simulated histograms of Myosin angular distribution the presence or absence of vorticity (rotation rates of $0°$/min and $3°$/min, myosin-effective lifetime of 5 min). Compare with histograms in (**h'**) and (**i**). (**i**) Junctional myosin in a ventro-posterior region of the embryo at 25 and 35 min post VF initiation, showing the recovery of myosin/DV axis alignment. Junctions colored according to their orientation and fluorescent intensity as in (**f**). Histogram shows the distribution of orientations observed at 25 and 35 min.

Next, we characterized the orientation of myosin anisotropy at the tissue scale. The angle between the local myosin anisotropy and the DV axis at every point of the germband over time is a 3D dataset that we broke down in several ways. We first analyzed the orientation of detected MRJs with respect to the DV axis as a function of time (*Figure 3b*). Consistent with the high autocorrelation of myosin orientation (*Figure 2f*), we found only a small change in the spatially averaged myosin/DV axis angle. Before the initiation of GBE, the myosin/DV axis angle was 2°. After the onset of GBE flow, measured at 10 min post VF initiation, the spatial average increased and reached a maximum of 8° at 25 min. Strikingly, after 25 min post VF initiation, the average myosin/DV axis angle began to decreases again, as the orientation of myosin anisotropy realigned with the DV axis. We refer to this phase of myosin behavior as the 'recovery phase'.

We also carried out a regional, stripe-specific analysis to account for the fact that the tissue flow field responsible for advection varies across the germband (*Figure 3c* and *Appendix 1—figure 9*). Since PRGs advect with cells, we used the angle of individual stripes with respect to the DV axis to characterize local tissue-level reorientation, allowing us to test the degree to which myosin orientation is affected by advection. For Runt stripe 5 (*Figure 3c*), the Runt/DV axis angle was approximately 3° at the beginning of GBE and then increased monotonously starting at 10 min, exceeding 20° by 30 min post VF formation. The myosin/DV axis angle measured in the region of stripe 5 also increased after 10 min. This increase was notably less than that of the Runt/DV axis angle from which it clearly diverges by 15 min. By 28 min, myosin deflection away from the DV axis reached a maximum of 12°, followed by a recovery phase. The same analysis for runt stripe 3 showed neither of the two quantities accumulated an angle with the DV axis that exceeded 4° (*Appendix 1—figure 9*).

The two stripes, 3 and 5, are in spatially distinct regions of the tissue flow field. Runt stripe 3 coincides with the central region of the flow, where instantaneous flow and thus cell trajectories were mainly parallel to the DV axis (*Figure 3d and e*). Consequently, we expected the local sense of orientation to be conserved. Runt stripe 5 was passing through a vortex that we expected to rotate cell junctions (*Figure 3d*). To test this idea, we first characterized the rate of local rotation in degrees per minute due to tissue flow (*Figure 3e*) by calculating the vorticity (*Figure 3e'*, *Landau and Lifshitz, 1987*). At the whole embryo level, the spatial pattern of the vorticity had a broad peak of about 5° per minute around the domain of the posterior vortex and a small peak below 2° per minute on the anterior, vanishing elsewhere. We next characterized the spatial pattern of the orientation of MRJs with respect to the DV axis (*Figure 3f*). Across most of the embryo surface, MRJs were parallel with the DV axis, with a clear exception in a domain around the posterior vortex, where individual junctions deflected as far as 20° from the DV axis (*Figure 3f'*). This spatial correlation between vorticity and DV axis deflection also extends across time, with a 5 min delay between the time course of vorticity and the myosin angle defect (*Figure 3b*).

## The interplay of subcellular and tissue-level dynamics quantitatively captures the myosin pattern

Next, we asked if the vorticity and the myosin/DV axis angle could be causally linked. Myosin stayed bound on junctions for an extended but finite amount of time (*Figure 3a'*). We postulated that myosin motors preferentially bind to junctions when they are parallel to the DV axis. Once these junctions are rotated by tissue flow, the myosin orientation is rotated with them, until motors begin to detach (*Figure 3d*). The resulting deflection angle is given by the product of rotation rate and myosin lifetime, which we reasoned could quantitatively account for the orientation defect of myosin in regions of high vorticity. We implemented this geometric source hypothesis in a mathematical model that describes the time course of the myosin concentration $m$ on a junction with orientation $\theta$:

$$\frac{d\theta}{dt} = \text{tissue rotation rate} \tag{2}$$

$$\frac{dm}{dt} = +\text{static source}(\theta) - \frac{m}{\tau} \tag{3}$$

The model assumes that (i) myosin binds to junctions that are parallel to the DV axis (the 'static source' is peaked around $\theta = 0$), (ii) myosin unbinds with rate $1/\tau$, and (iii) that junctions rotate proportional to local vorticity. It is also possible to consider a model where the myosin recruitment rate is constant and, instead, the detachment rate is modulated according to junction orientation. We find that such a model cannot account for the observed deflection of myosin orientation (see Appendix, 'Effects of modification of myosin lifetime by static source'). Further, in addition to tissue rotation, junctions could in principle be rotated by tissue strain. This is however not the case in the germ band because the majority of tissue strain is due to cell rearrangement. Junction shortening or lengthening also does not affect junction orientation. For a detailed discussion, see the Appendix , 'Additional effects in static source model', where we also discuss the influence of tissue curvature on junction rearrangement . Finally, related ideas have previously been proposed in *Farrell et al., 2017* on a qualitative level.

Equation 2 for a single junction can be transformed to describe the entire angular distribution of junctional myosin (see Appendix I.7) and used to predict the local average deflection angle $\bar{\theta}$ of the myosin anisotropy:

$$\frac{d\bar{\theta}}{dt} = \text{tissue rotation rate} - \frac{\bar{\theta}}{\tau} \tag{4}$$

Crucially, $\bar{\theta}$ only depends on the direction around which recruitment is maximal and is independent of its magnitude, potential spatial modulation, and functional form (see Appendix I.7), in agreement with the reasoning presented above. The only free parameter is the effective myosin lifetime $\tau$, which captures the duration that myosin motors remain bound on junctions (see Appendix I.7). Our model makes several quantitative predictions about junction dynamics that we tested (*Figure 3b–c and e–i*).

First, we computed the predicted angle between the orientation of myosin anisotropy and the DV axis by solving *Equation 4* (see Appendix for mathematical definition). We assume that the parameter $\tau$ is constant in time and across the embryo. Using an ensemble of $N = 5$ embryos, we then fitted the parameter $\tau$ by minimizing the average difference between the myosin orientation predicted from *Equation 4* and the observed orientation across the germband during convergent extension. We refer to the angle between the vorticity-based prediction and the actual myosin angle as the 'rotation-rate adjusted' myosin/DV axis angle. We found that the vorticity-adjusted junction orientation aligned well with the DV axis when the duration $\tau$ of myosin binding was 5 min (*Figure 3c and g*). Throughout GBE – even in regions of highest vorticity – the rotation-rate-adjusted angle remained close to 0° , while the discrepancy between the Runt and myosin orientations continuously increased (*Figure 3c* and *Appendix 1—figure 9*). *Figure 3b* shows the predicted spatial average myosin/DV axis angle (see Appendix I.7.1), which is in good agreement with the observed data.

The value of $\tau$ found by our fit is qualitatively similar, but somewhat larger, than the FRAP-measured myosin lifetime. As discussed in the Appendix, this is because FRAP measures the time individual motors remain on a junction, while $\tau$ measures how long the total myosin level on a junction persists. The latter time can be larger if factors that affect myosin levels, such as kinases, are longer-lived than individual motors, or if there is a positive feedback of current myosin levels on myosin recruitment.

Second, we analyzed the angular distribution of MRJs, i.e., the range of orientations of MRJs detected in a small tissue patch, and measured its spread (*Figure 3h–h"* and *Appendix 1—figure 16*). The dynamics of the angular distribution depended on location within the embryo. The standard deviation remained nearly constant in regions of low vorticity, for example, around Runt stripe 3 (*Figure 3h*, *Figure 2h–h'*). In regions of high vorticity, the standard deviation rapidly changed, giving rise to a much broader distribution (*Figure 3h'*, *Figure 2i–i'*). This feature is accurately captured in our model: without vorticity there will be no reorientation, and only junctions parallel to the DV axis will recruit myosin (*Figure 3h"*). With vorticity, junctions that recruited myosin while aligned with the DV axis will rotate. Since myosin stays bound for a extended but finite lifetime, the distribution widens (*Figure 3h"*).

Third, we analyzed the time course of vorticity. It increased with the onset of GBE, reached a plateau by 20 min, and slowed down by 25 min (*Figure 3b*). Strikingly, MRJs realigned with the DV axis once vorticity decreased (*Figure 3b and i*), as predicted by *Equation 3*. The time delay between vorticity and deflection (*Figure 3b*) is expected from our model: it takes time to deflect junctions in response to vorticity and, correspondingly, for myosin to detach from deflected junctions once vorticity decreases. At 25 min, MRJs in a posterior region on the ventrolateral side of the embryo were strongly deflected, with a median angle of about 20° to the DV axis (*Figure 3i*). 10 min later, the median of the distribution shifted by 21° , aligning with the DV axis. The dynamics of the distribution during recovery – i.e., while vorticity decreases – mirrors the dynamics during the onset of vorticity, supporting the notion that the myosin angular distribution is mainly determined by the strength of the local vorticity. Indeed the simulation in *Figure 3h"* matches the observations in both *Figure 3h'* and *Figure 3i* (of course, the magnitude of vorticity change in *Figure 3i and h'* is different, leading to quantitative differences in the myosin orientation). These observations were consistent with the geometric source hypothesis, both qualitatively and quantitatively (*Figure 3c*).

Our mathematical model accounts for the dynamics of myosin orientations in terms of the rotation due to flow, and the extended but finite binding lifetime of myosin motors to junctions. The model accurately describes key features of the spatiotemporal dynamics of the mean as well as standard deviation of the distribution of MRJ orientations. The model's single parameter is the same across the embryo, constant in time, and agrees qualitatively with recovery kinetics as measured on individual junctions experimentally. Our global, comprehensive analysis confirms and extends previous local, tracking-based observations that showed that junctions can remodel their myosin levels as they rotate (*Farrell et al., 2017*). These results point towards control of myosin orientation by static geometric cues, as opposed to the passively advected PRGs and Tartan stripes.

## Myosin dynamics in patterning and geometric mutants confirms static orientation of myosin recruitment

Harnessing the genetic toolkit available in the *Drosophila* model system, we show that we can account for the behavior of mutants by modulating parameters of our mathematical model – vorticity, myosin kinetics, and geometry (*Figure 4a–a"*). Twist is expressed in the VF and *twist*[ey53] mutants have a defect in VF formation, accompanied by reduced kinetics across the entire embryo (*Martin et al., 2009*; *Butler et al., 2009*; *Streichan et al., 2018*; *Gustafson et al., 2021*). We found the average speed of GBE in *twist*[ey53] mutants was reduced by a factor of two compared to WT (*Figure 4b*). This was accompanied by a corresponding reduction of the vorticity (*Figure 4a and c*). The myosin/DV axis angle was likewise smaller (*Figure 4a and c*). Fitting the model to this data revealed a similar myosin binding lifetime as in WT, and the rotation rate-adjusted angle of MRJs closely aligned with the DV axis.

Analysis of fixed samples has demonstrated that during GBE, myosin polarization in PRG and TLR mutants is reduced (*Paré et al., 2014*). GBE tissue flow in these mutants is impaired as well, particularly in later phases (*Irvine and Wieschaus, 1994*). Our dynamic data indicates that myosin and PRGs/TLRs are regulated in different frames of reference. This raises the question of how patterning gene expression can be quantitatively linked to myosin anisotropy and – by extension – tissue flow. To study this, we performed live imaging of myosin in *eve*[R13] mutants (*Figure 4a', b, d and e*). We found that the initial kinetics of GBE closely match that of WT embryos. However, at 10 min post VF formation the kinetics of *eve*[R13] mutants change abruptly (*Figure 4b*). Additionally, vorticity was reduced compared to WT (*Figure 4d*). We detected anisotropic myosin in the germband during GBE (*Figure 4e*), although anisotropy was significantly reduced in comparison to WT (*Appendix 1—figure 18*). We found that

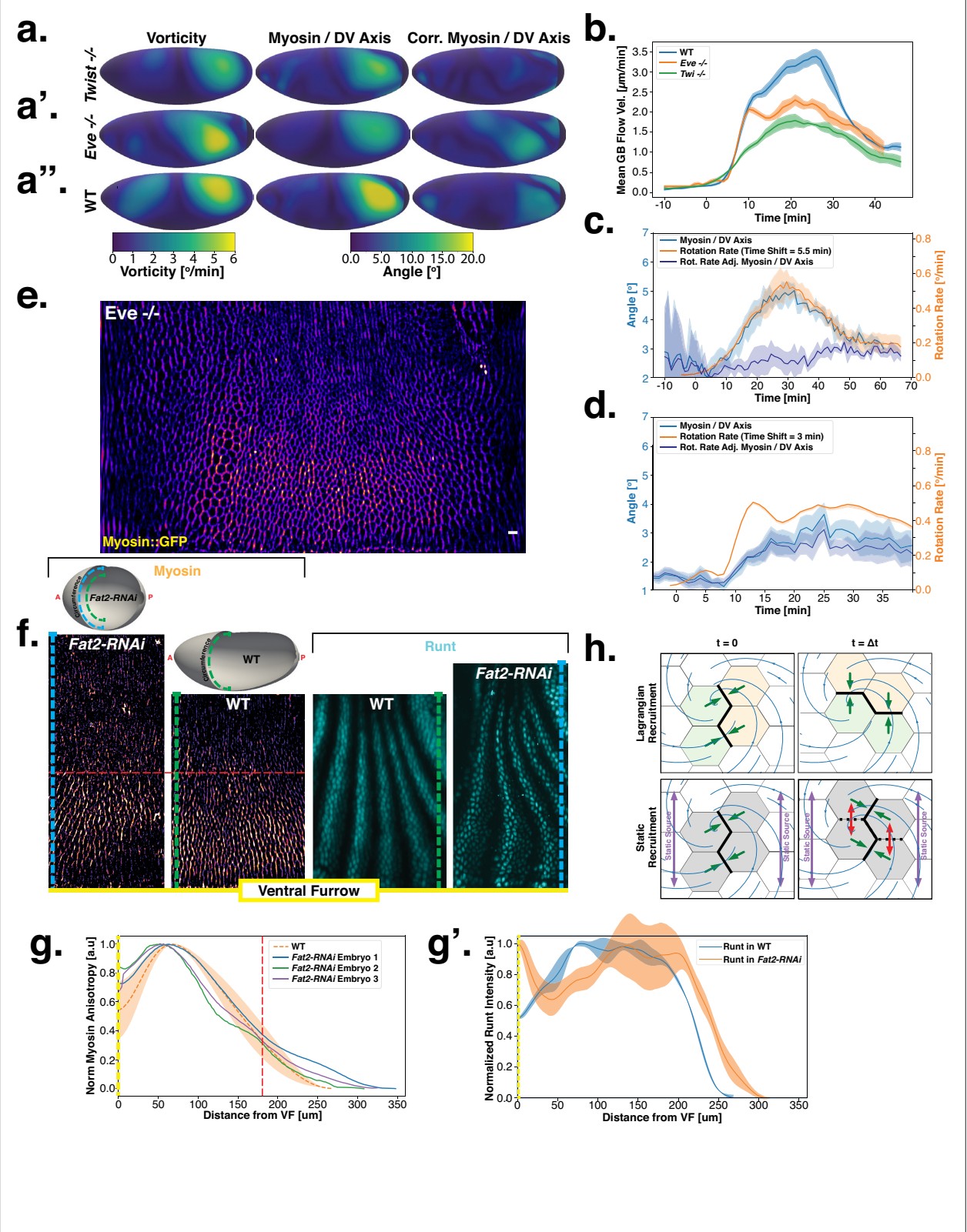

**Figure 4.** Dynamics of myosin orientation in mutants affecting vorticity or embryo geometry can be quantitatively described. (**A–A″**) Vorticity, myosin/dorsal-ventral (DV) axis angle, and rotation-rate adjusted myosin/DV axis angle prediction in $N = 5$ WT, $N = 4$ $twist^{ey53}$, and $N = 3$ $eve^{R13}$ embryos. Heatmaps show a temporal average from 15 to 25 min post ventral furrow (VF) initiation. The correlation of vorticity and myosin/DV axis angle persists in all mutants, and the rotation rate-corrected myosin/DV axis angle remains low. (**B**) Spatial average of tissue flow velocity in the germband

*Figure 4 continued on next page*

*Figure 4 continued*

over time in $N = 5$ WT, $N = 4$ *twist*$^{ey53}$, and $N = 3$ *eve*$^{R13}$ embryos. *twist*$^{ey53}$ and *eve*$^{R13}$ mutants show markedly lower flow velocity than WT, but differ in their kinetics. (**C**) Spatial average of vorticity, myosin/DV axis angle, and rotation rate-adjusted myosin/DV axis angle over time in $N = 4$ *twist*$^{ey53}$, Myosin::GFP embryos. (**D**) Spatial average of vorticity, myosin/DV axis angle, and rotation rate-adjusted myosin/DV axis angle over time in $N = 3$ *eve*$^{R13}$, Myosin:mCherry embryos. (**E**) Junctional myosin in *eve*$^{R13}$ mutants remains anisotropic and aligned with the DV axis, although the degree of anisotropy is reduced. One lateral half of a representative *eve*$^{R13}$, Myosin::mCherry embryo, 18 min post VF initiation. (**F**) Top: 3D shape of WT embryos and embryos from *Fat2-RNAi* mothers, extracted by tissue cartography pipeline. Compared to WT, *Fat2-RNAi* embryos are spherical and have a greatly increased circumference (marked in green resp. blue). Bottom: patterns of junctional myosin and Runt in the germband of WT and *Fat2-RNAi* embryos at equivalent phases in germband extension (GBE) (10 min post VF initiation in WT). Only one lateral half is shown, orientation is dorsal up, anterior left. Junctional myosin is visible up to the same distance from the VF in both WT and *Fat2-RNAi* embryos. (**G–G′**) Quantification of the decay of junctional myosin (**G**) and Runt (**G′**) away from the ventral furrow in WT and *Fat2-RNAi*. Myosin data from $N = 5$ WT and $N = 3$ *Fat2-RNAi* embryos. Runt data from both lateral halves of WT and *Fat2-RNAi* embryo shown in (**F**). (**H**) Myosin recruitment by a passively advected source vs by a static source leads to qualitatively and quantitatively different behavior.

MRJs were mainly aligned with the DV axis, except in the region of the posterior vortex (*Figure 4a′*). The best fit of the static source model to this data suggests that myosin binding lifetime is significantly reduced in *eve*$^{R13}$ mutants (to $\tau = 2 - 3\text{min}$). We therefore carried out FRAP experiments on junctional myosin in *eve*$^{R13}$ mutants (*Appendix 1—figure 11*). The FRAP data shows complex modifications to the myosin dynamics, but is compatible with overall more rapid myosin dynamics in *eve*$^{R13}$ compared to WT embryos (see *Appendix 1—figure 11*). A joint reduction of vorticity, and myosin lifetime can account for the near constant time course of the MRJs angle defect with the DV axis (*Figure 4d*).

Finally, by knocking down the atypical cadherin Fat2 (*Fat2-RNAi*) in somatic ovarial cells in female flies, we created nearly spherical embryos (*Chan et al., 2019*) with a shorter, but variable length, and up to 30% extended DV circumference (*Figure 4f*, top). We originally planned to use *Fat2-RNAi* to modify the direction of the proposed static source, the geometric DV axis. However, since *Fat2-RNAi* eggs remain highly rationally symmetric about the AP axis, the direction of the geometric DV axis is not significantly affected. Instead, this mutant provided an opportunity to test if PRG stripes would change in the same way as MRJs around the ectopically extended DV circumference. The AP patterning system remains intact in *Fat2-RNAi* embryos (*Chan et al., 2019*): PRGs were expressed in seven stripes along the AP axis, with a decrease in expression at the dorsal pole (*Figure 4f*, right). Similar to WT, myosin recruitment to junctions was strongest in ventral regions and dropped markedly on the lateral side. In both WT and *Fat2-RNAi* embryos, MRJs were detected up to ~175 μm away from the VF in the lateral ectoderm (*Figure 4g–g′*). Strikingly, since the absolute length of the DV circumference is larger in *Fat2-RNAi* embryos than it is in WT embryos, and the PRG stripes extended normally to the dorsal pole, there is a substantial region on the lateral surface of *Fat2-RNAi* embryos where PRG stripes are clearly visible but no myosin anisotropy could be detected (*Figure 4f*). The magnitude of tissue flow in *Fat2-RNAi* was reduced (*Appendix 1—figure 19*). Moreover, MRJ orientations changed little due to low spatial overlap between the regions of high vorticity and myosin recruitment (*Appendix 1—figure 20*). These observations suggest that presence of striped PRG expression is not sufficient to set up myosin anisotropy.

One possibility is that PRGs play a role in directing myosin anisotropy in an initial phase, with control over myosin orientation transferred to a static cue once flow starts. However, this hypothesis is not supported by the lack of linear correlation between PRGs and the initial myosin pattern, the results in *Fat2-RNAi* embryos, and the results of *Gustafson et al., 2021*, which found that mechanical cues can explain the early myosin pattern. Taken together, our results suggest that instead of directly instructing anisotropic myosin recruitment, PRGs might influence the myosin anisotropy by regulating retention of myosin to junctions.

## Discussion

Here, we presented a quantitative study dissecting the dynamic rules governing myosin anisotropy during *Drosophila* GBE. We found that the orientation of MRJs closely tracks the DV axis, a static geometric landmark. By contrast, the localization of patterning genes (PRGs and Tartan) implicated in GBE deform due to advection with the flowing tissue and deflect away from the DV axis over time (*Figure 4h*). We define a mathematical model that accounts for the dynamics of myosin orientation as

a product of tissue flow vorticity and the extended-but-finite time that myosin motors remain bound to junctions.

These results suggest that the known upstream regulatory factors of GBE – PRGs and TLRs – are passively advected in qualitative difference compared to the nearly-static myosin pattern. This observation further highlights the complex nonlinear nature of the hypothesized instructive link between anisotropic myosin recruitment and local differences in PRG levels between adjacent cells. Results from *Fat2-RNAi* embryos further indicate that the presence of PRG stripes is not sufficient for anisotropic myosin recruitment. Our model presents a simpler alternative with only a single parameter and suggests a clearly interpretable biophysical role for PRGs (likely via the TLRs): modulating the sensitivity of myosin recruitment to the static source and regulating myosin maintenance on junctions.

Our dynamic data from WT as well as multiple mutant genotypes are consistent with preferential myosin recruitment along the DV axis. However, the mechanism underlying myosin recruitment remains unclear. Myosin dynamics can be organized not only by instructive genetic signals but also by mechanical inputs (*Fernandez-Gonzalez et al., 2009*; *Petridou et al., 2017*; *Gustafson et al., 2021*). Crucially, mechanical cues such as epithelial tension are not necessarily advected by tissue flow.

Strain-responsive myosin recruitment, triggered by DV strain due to the invagination of the VF, establishes early myosin anisotropy, acting as starting signal for GBE (*Gustafson et al., 2021*). Yet VF formation is transient, raising the question of how anisotropic recruitment is maintained during later stages of GBE. One stationary signal with the required anisotropy is mechanical feedback triggered by epithelial stress (*Rauzi et al., 2010*; *Munjal et al., 2015*; *Noll et al., 2020*). The static stress anisotropy might originate from turgor pressure within the embryo (*Lu et al., 2016*), which the surface stress needs to balance (*Noll et al., 2017*). Due to the cylinder-like geometry of the embryo, this results in a static, anisotropic surface stress ('hoop stress') (*Audoly and Pomeau, 2010*). Cortical tension due to turgor pressure is known to play a crucial role in mouse blastocyst development (*Chan et al., 2019*). Tools for faithful measurement and manipulation of hoop stress will be needed to further evaluate this hypothesis.

The geometric control of myosin orientation described here has close parallels to primitive streak formation in the early quail embryo (*Caldarelli et al., 2021*) as well as to other model processes of convergent extension, the *Drosophila* wing disc (*Aigouy et al., 2010*) and the *Xenopus* larval epithelium (*Chien et al., 2015*). In the the latter two systems, planar cell polarity proteins orient according to mechanical inputs propagated over tissue-length scales. Our results suggest that underlying biological complexity notwithstanding, the dynamics of morphogenesis can be quantitatively described by simple models with few and clearly interpretable parameters.

## Materials and methods

### Lightsheet microscopy

#### Microscopy

Lightsheet data sets were taken on a custom Multi View Selective Plane Illumination Microscope (MuVi SPIM) (*Krzic et al., 2012*) with scatter reduction through confocal imaging (*de Medeiros et al., 2015*). This microscope is capable of fluorescent imaging of the entire *D. melanogaster* embryo at subcellular resolution and was previously described in detail in *Gustafson et al., 2021*. Electronics were controlled using MicroManager (*Edelstein et al., 2014*).

#### Image acquisition

Prior to imaging, embryos were dechorionated and mounted in low-melting point agarose gel (*Krzic et al., 2012*). Samples are imaged simultaneously by two objectives at opposite sides of the embryo, with lighsheet $z$-sections spaced by 1.5 $\mu$m. By rotating the embryo by $45°, 90°$, and $135°$, and repeating the $z$-imaging, we create a total of eight views per time point that are registered and fused to create a volumetric dataset in the next step. All lightsheet movies in this work are taken at a time resolution of 1 min.

### Data fusion and surface extraction

Images recorded by the lightsheet microscope were registered based on the position of fiduciary beads embedded in the agarose (Fluoresbrite multifluorescent 0.5 μm beads 24054, Polysciences Inc,

as described in *Gustafson et al., 2021*) using the Multiview reconstruction plugin (*Preibisch et al., 2014*) in Fiji (*Schindelin et al., 2012*). We used all-to-all registration, mapping all perspectives at all time points to a common reference frame using an affine transformation. Images were then deconvolved and fused using the algorithm introduced in *Preibisch et al., 2014*, yielding images with an isotropic resolution of 0.2619 µm.

The embryo surface is detected within the resulting volumetric data using an Ilastik detector (*Berg et al., 2019*), to which a surface was fitted using the ImSAnE software (*Heemskerk and Streichan, 2015*), which was used for tissue cartography as described in *Heemskerk and Streichan, 2015*. To improve accuracy, we applied two iterations of the Ilastik + ImSAnE workflow. The resulting 'onion' layers normal to the embryo surface, spaced 1.5 µm, were used to generate maximum-intensity projections.

## Confocal imaging and FRAP

Imaging for the FRAP experiments shown in **Figure 3** and the $eve^{R13}$ data shown in **Appendix 1—figure 17** was done using a Leica SP5 confocal microscope and ×63/1.4 NA oil immersion objective at a frame rate of one frame per 1.78 s. Junctions were tracked manually using Fiji (*Schindelin et al., 2012*). We bleached regions of size $5\mu m \times 5\mu m$ for approximately 8 s using 50 mW laser power.

## Fly stocks and genetics

A full stock list is presented in Table S1. The fluorescent fusion proteins used in this study include Myosin::GFP (II or III, sqh::GFP, *Royou et al., 2002*), Myosin::mCherry (II, sqh::mCherry, *Martin et al., 2009*), Runt::LlamaTag-GFP (*Bothma et al., 2018*, gift from H. Garcia), Eve::YFP (III, *Ludwig et al., 2011*), Gap43::mCherry (III, Membrane::mCherry, *Martin et al., 2010*).

Recombinant chromosomes containing the chromosomal deficiency Df(2L)dpp[s7-dp35] 21F1–3;22F1–2 (*halo*) and either $eve^{R13}$, or $twist^{ey53}$ were balanced with CyO. The chromosome containing $eve^{R13}$ was recombined with Myosin::mCherry (II). *Halo, twist*$^{ey53}$ embryos were balanced with a version of CyO that also contains Myosin::GFP. Homozygous $eve^{R13}$ and $twist^{ey53}$ embryos were identified based on visualization of the *halo* phenotype while heterozygous control embryos did not show the *halo* phenotype.

The following stocks were used to generate reduced aspect ratio (*Fat2-RNAi*) embryos: w; Traffic jam-Gal4; Myosin::GFP; Gap43::mCherry, w; Myosin::GFP; UAS-*Fat2-RNAi* (*Chanet et al., 2017*).

## Immunohistochemistry and antibody production

For heat fixation, embryos were dechorionated with 50% bleach and then fixed using heat and methanol as described previously (*Müller and Wieschaus, 1996*). Primary antibodies for immunohistochemistry were Runt (guinea pig, 1:500, Wieschaus Lab), Even-Skipped (rabbit, 1:500, gift from M.Biggin), Fushi-Tarazu (rabbit, 1:1000, gift from M.Biggin), Paired (mouse, 1:100, gift from N.Patel), Sloppy-Paired (rabbit, 1:500, gift from M.Biggin), Hairy (rat, 1:100, Wieschaus Lab), and Tartan (rabbit, 1:100, this study, GenScript, based on full-length peptide). Donkey and goat secondary antibodies conjugated to Alexa Fluor 488, 561, and 647 were used (1:500, Thermo Fisher Scientific). Embryos were mounted in 1.5% low gelling temperature agarose (Millipore Sigma-Aldrich) for light-sheet imaging, and mounted in 50% PBST 50% Aqua-Poly/Mount (Polysciences) for confocal imaging.

## Image processing and analysis software

Image processing, described in detail in the SI, used a combination of custom Python scripts using the Scientific Python (*Virtanen et al., 2020*) and scikit-image (*van der Walt et al., 2014*) packages and custom MATLAB scripts. These scripts are available at *Claussen and Streichan, 2022*. Tissue cartography was performed using the ImSAnE software (*Heemskerk and Streichan, 2015*). Surface detection, cell tracking, and segmentation of Runt stripes were performed using Ilastik (*Berg et al., 2019*).

## Acknowledgements

The authors thank Eric Wieschaus, Boris Shraiman, Fridtjof Brauns, and members of the Streichan lab for valuable discussions and suggestions. We additionally wish to thank Eric Wieschaus for providing several fly lines, Sophie Streichan for aid in handling stocks, crosses, and reagents, and Cécile Regis for assistance with 3D visualizations. This research was supported by NIH grant no. R35 GM138203 and

partially supported by the National Science Foundation under grants PHY-1707973 and PHY-1748958. MFL acknowledges support by NIH F31 fellowship no. HD093377. NPM acknowledges support from the Helen Hay Whitney Foundation.

## Additional information

### Funding

| Funder | Grant reference number | Author |
|---|---|---|
| National Institutes of Health | 5 R35 GM138203 | Sebastian J Streichan |

The funders had no role in study design, data collection and interpretation, or the decision to submit the work for publication.

### Author contributions

Matthew F Lefebvre, Data curation, Formal analysis, Validation, Investigation, Visualization, Methodology, Writing – original draft, Writing – review and editing; Nikolas H Claussen, Data curation, Software, Formal analysis, Validation, Investigation, Visualization, Methodology, Writing – original draft, Writing – review and editing; Noah P Mitchell, Data curation, Software; Hannah J Gustafson, Data curation, Methodology; Sebastian J Streichan, Conceptualization, Resources, Software, Supervision, Funding acquisition, Validation, Investigation, Methodology, Writing – original draft, Project administration, Writing – review and editing

### Author ORCIDs

Matthew F Lefebvre ⬤ http://orcid.org/0000-0002-9590-4293
Nikolas H Claussen ⬤ http://orcid.org/0000-0002-9020-6437
Sebastian J Streichan ⬤ http://orcid.org/0000-0002-6105-9087

### Decision letter and Author response

Decision letter https://doi.org/10.7554/eLife.78787.sa1
Author response https://doi.org/10.7554/eLife.78787.sa2

## Additional files

### Supplementary files
• MDAR checklist

### Data availability

All data for this article is available publicly without any restrictions. In our article, we make use of two datasets: (1) confocal microscopy data of FRAP experiments, which is available on the Dryad repository https://doi.org/10.25349/D94C8M; (2) lightsheet microscopy data of entire embryos. The data we use in the current publication is a subset of a larger dataset, the 'Morphodynamic atlas of *Drosophila* development', which is publicly available on the Dryad repository https://doi.org/10.25349/D9WW43. This collection is indexed by the fly genotype and fluorescent marker imaged, so that the movies and images used in the current publication can be found easily. Lightsheet microscopy integrates microscopy and computational processing and its computational pipeline creates intermediate, 'raw' data files, which are of very large size (TBs for a single movie). This raw data is available upon request from the corresponding author without restriction or need for a specific research proposal. The analysis code used is available on GitHub https://github.com/nikolas-claussen/Geometric-control-of-Myosin-II-orientation-during-axis-elongation (*Claussen and Streichan, 2022*; copy archived at swh:1:rev:2e8118a1f0e56a4a402ff73c1c8a206f8f8605e9).

The following datasets were generated:

| Author(s) | Year | Dataset title | Dataset URL | Database and Identifier |
|---|---|---|---|---|
| Lefbvre M, Claussen N | 2022 | Early *Drosophila* Spaghetti-Squash-GFP FRAP | https://doi.org/10.25349/D94C8M | Dryad Digital Repository, 10.25349/D94C8M |
| Mitchell N, Lefebvre M, Jain-Sharma V, Claussen N, Raich M, Gustafson H, Bausch A, Streichan S | 2022 | Morphodynamic atlas for *Drosophila* development | https://doi.org/10.25349/D9WW43 | Dryad Digital Repository, 10.25349/D9WW43 |

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

# Appendix 1

## Quantitative image analysis

### Tissue cartography and image analysis on curved surfaces

To analyze the in toto 3D data obtained by light-sheet microscopy (*Krzic et al., 2012*), we use tissue cartography as described in *Heemskerk and Streichan, 2015*. Briefly, the embryo surface is detected using a machine-learning pixel classification workflow and a smooth surface is fit to the resulting point cloud. This surface is refined in a second step, repeating classification and fitting. This surface can be evolved along its normal to create so-called onion layers. We created pullbacks showing the pixel intensity on these onion layers and obtained the final image by a maximum projection along across layers.

The result shows the embryo in a cylindrical chart with anterior left and posterior right. The ventral midline corresponds to the cut used to unroll the cylinder. Velocity fields, gradients of fluorescent intensity, and orientations of cell edges can then be computed within this chart. All angles are calculated with respect to the induced metric, correcting for distortions induced by the cylindrical projection near the poles.

For *Fat2-RNAi* embryos, which have a particularly round shape, we also used a second cylindrical projection, analogous to the Mercator projection of the earth, which is angle-preserving at all points. This allows our edge-detection algorithm (see below) to work faithfully and avoids spurious anisotropy detection near the poles. The detected edges were then mapped back to the original cylindrical chart.

### Time alignment and dynamic atlas

In our work, we combine and compare data from different embryos carrying with complementary makers. Note that because we employ in toto imaging and tissue cartography, spatial alignment is trivial. Temporal alignment is facilitated by the extreme reproducibility of the movements of gastrulation across embryos. Embryos were time-aligned using two methods.

### Time alignment of fixed embryos

Fixed embryos were time-aligned using a landmark-based approach. In addition to its primary staining, each fixed embryo way stained for Runt, and the anterior boundary of the seventh Runt stripe was computationally extracted. The same was done for $N = 5$ live recordings of embryos with fluorescent tagged Runt. Live movies were time-aligned to one another using all-to-all optimization of the similarity of the extracted stripe contour across movies. The stained samples were aligned to the resulting master time line. The details of this 'dynamic atlas' method will be described in a forthcoming paper (*Mitchell et al., 2022*).

### Time alignment of live data

To time-align different live films not tagged for Runt, we compared their PIV-calculated flow fields as in *Streichan et al., 2018*. PIV fields were calculated using the phase-correlation method as in *Streichan et al., 2018*. For each genotype, we chose one reference movie to which the remaining movies are aligned with a constant time shift $t_{off}$, obtained by minimizing the average difference of the velocity fields:

$$t_{off} = \operatorname{argmin}_{t'} \int dt \int d^2x \, \|\mathbf{v}(t + t') - \mathbf{v}_{ref}(t)\| \tag{5}$$

Here, $\mathbf{v}$ is the velocity field of the embryo to be aligned, $\mathbf{v}_{ref}$ that of the reference movie, and the integral represent averages over time and the embryo surface. *Appendix 1—figure 1* shows the aligned the spatial average velocities of $N = 5$ WT Myosin::GFP embryos over time. *Appendix 1—figure 1* highlights that time-alignment is possible within $\pm 1\mathrm{min}$ due to the highly reproducible time courses.

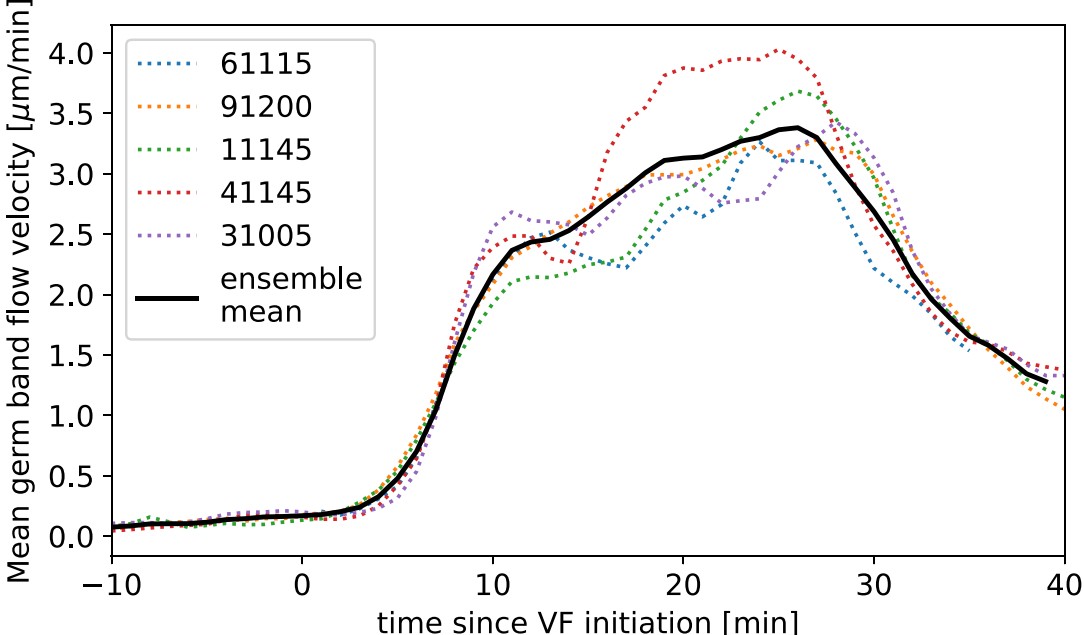

**Appendix 1—figure 1.** Time-alignment using particle image velocimetry (PIV) curves.
Average flow velocity in germband during germband extension (GBE) in $N = 5$ WT Myosin::GFP, time-aligned by matching PIV fields.

## Ensemble averages and meso-scale analysis

Using the time-alignment obtained, we can compute ensemble-averages across embryos, such as the average velocity field or the average myosin anisotropy. The ability to faithfully compute ensemble averages and thus distill the behavior of the stereotypical embryo and make statistically significant statements is a key advantage of our workflow that combine in toto tissue cartography, time-alignment, and mesoscale analysis.

In meso-scale analysis, we smooth cell-level quantities over the scale of ~3–5 cells to obtain everywhere-defined tissue-level fields (*Streichan et al., 2018*). This process drastically reduces noise, focuses on the tissue-level dynamics relevant for large-scale morphogenesis, allows easy comparison across embryos, and defines suitable inputs for the type of quantitative, predictive model we study in this rticle. Examples of this approach are the local myosin anisotropy orientation, the smoothed PRG gradients, and the PIV-computed tissue flow field.

We use the ensemble velocity field of $N = 5$ WT Myosin::GFP embryos for the calculation of the vorticity in *Figure 3* and for the transport of Runt stripes in *Appendix 1—figure 2*. To time-align Runt and myosin data, we align the Runt PIV fields to the myosin ensemble field according to the onset of GBE flow.

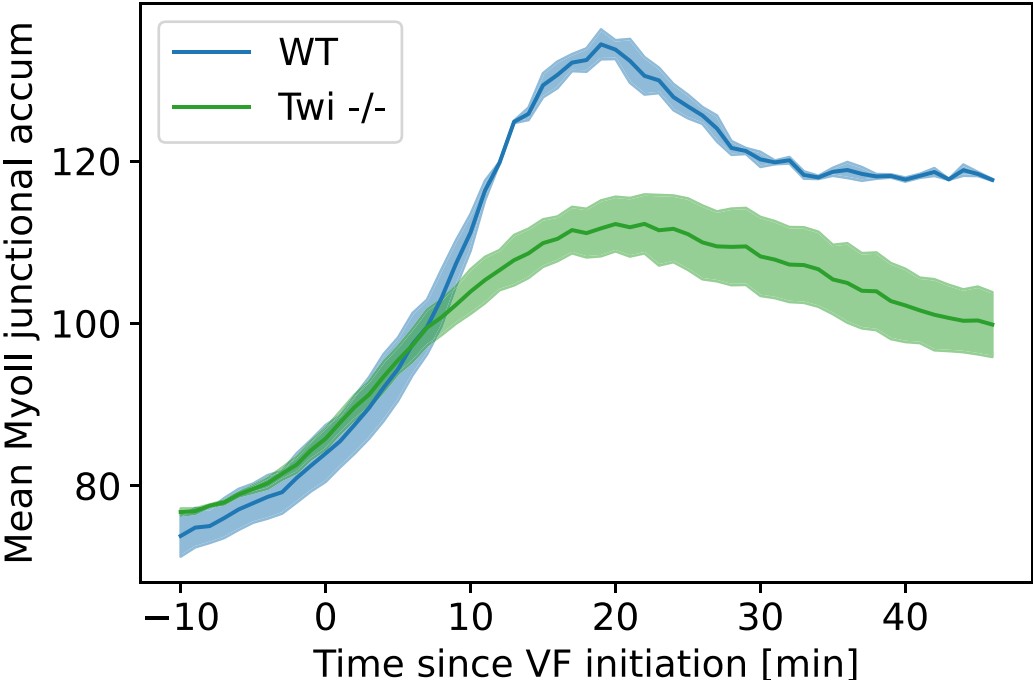

**Appendix 1—figure 2.** Average junctional myosin levels in WT and $twist^{ey53}$ embryos. Average junctional myosin levels, computed using the cytosolic normalization filter, in germband during germband extension (GBE) in $N = 5$ WT Myosin::GFP and $N = 4$ $twist^{ey53}$ Myosin::GFP embryos. Included in the spatial average are image regions with a junctional accumulation level of $\geq .05$, which we classify as 'junction.'

## Quantitative analysis of junctional myosin

### Cytosolic normalization of junctional myosin

We measure the concentration of fluorescent myosin motors in reference to the concentration of motors in the cytoplasm, an approach validated in *Gustafson et al., 2021*. The cytoplasmic intensity $I_c$ is calculated by applying a top-hat transform with a disk-shaped structuring element with a one-cell-diameter radius to the image data, intensity $I$. The cytosolically normalized signal is

$$I_n = \frac{I - I_c}{I_c}. \tag{6}$$

This measure is in principle independent of the concentration of fluorescently tagged molecules and allows to compare data from fly lines with different fluorescent tags. The cytoplasm acts as a pool from which motors can be recruited to the actomyosin cortex. If no motors have been recruited, the concentration near the membrane and in the cytosol will be equal, and the normalized signal, which measures excess junctional accumulation, vanishes.

*Figure 2* shows the average junctional myosin level on junctions in WT and $twist^{ey53}$ mutants over time. As previously reported (*Gustafson et al., 2021*), junctional myosin levels in $twist^{ey53}$ embryos are much lower than in WT due to slower recruitment during early GBE. This is a consequence of the lack of VF invagination in which generates forces that drive mechanosensitive myosin recruitment in WT.

### Segmentation-free edge detection by radon transform and computation of local myosin orientation

Cell segmentation at the whole-embryo scale is time consuming. Additionally, for anisotropically distributed markers such as myosin, it is often difficult detect cell edges with low marker levels and obtain correct cell outlines. We therefore used a segmentation-free technique previously presented and validated in *Streichan et al., 2018*. Briefly, the image is scanned with a local edge detection filter that analyzes circular patches of approximately one-cell diameter at a time. The filter applies the radon transform that computes the normalized line integral of the signal as a function of line

orientation $0 \leq \theta \leq 180°$ and offset of the line from the image center. In this way, edges are mapped to peaks in radon plane whose angle and offset correspond to edge orientation and real-space position and whose heights to the average image intensity along the edge. Such peaks can be detected robustly with different methods. Here, we consider only the global maximum in the radon plane since there is typically only a single edge in the filter window, additionally subject to a criterion filtering out insignificant peaks (peak elevation greater 1.5× the average). Alternative methods, e.g., the h-maxima transform, lead to equivalent results.

This method results in a list of edges containing their positions, orientations, and intensities. From this, we can compute the local average myosin anisotropy orientation, as well as the standard deviation of the orientation distribution. Since angles are defined only modulo 180° (e.g., an edge of angle 5° and 175° are in fact very close), directly averaging the angles can lead to distorted results. We therefore defined a nematic tensor $Q_e$ for each edge (*Doostmohammadi et al., 2018*):

$$Q_e = m_e \mathbf{n} \cdot \mathbf{n}^T \tag{7}$$

where $m_e$ is the average fluorescent intensity on the edge, and $\mathbf{n}$ is a unit vector parallel to the edge. These tensors can be locally averaged over a scale of $\sigma \approx 5$ cells to produce a tensor $Q_m$ representing the local myosin anisotropy. The top eigenvector of $Q_m$ defines the local myosin orientation used in *Appendix 1—figures 2–4*, and the top eigenvalue is the average magnitude of junctional myosin on edges aligned with the local orientation.

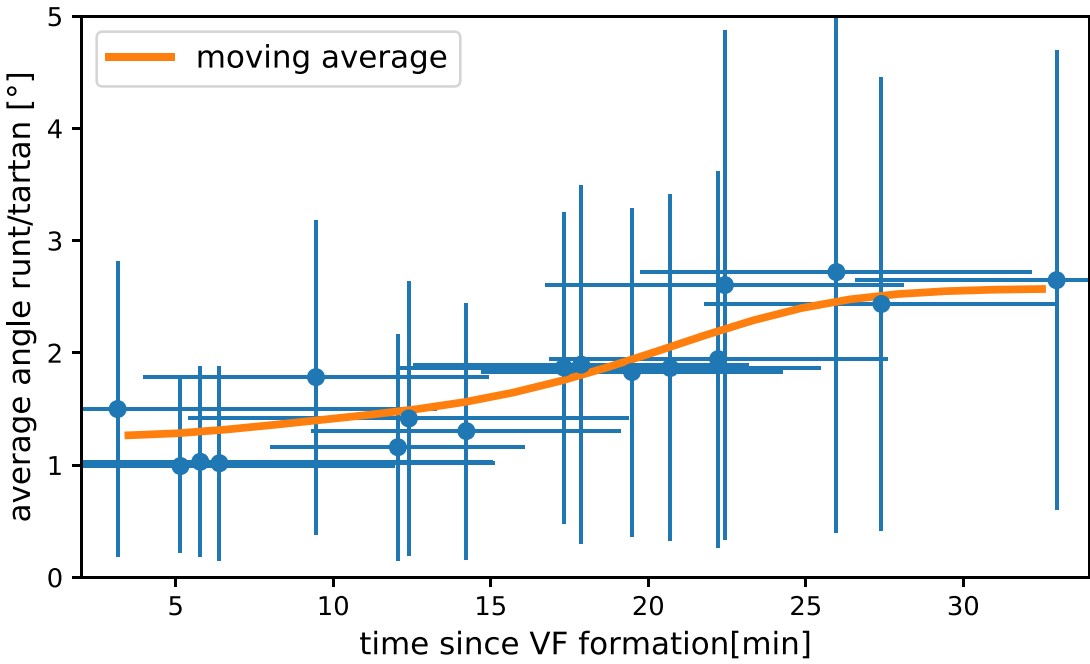

**Appendix 1—figure 3.** Measured angle between Runt and Tartan stripes. Runt angle from $N = 5$ Runt::LLamaTag-GFP embryos, Tartan data from $N = 17$ stained embryos.

## Definition of correlation coefficient for nematic fields

For nematic fields, such as the myosin tensor *Equation 6* and the Runt line field, the orientation angle is defined only up to 180°. We therefore use the following definition for the correlation between two nematic directors $\mathbf{n}, \mathbf{m}$, in particular in the correlation matrices *Figure 2B*:

$$\text{Correlation coefficient} = 2(\mathbf{n} \cdot \mathbf{m})^2 - 1 = \cos(2\theta) \tag{8}$$

where $\theta$ is the angle between the directors.

## Definition of Runt stripe angle

The local orientation of the Runt stripes was calculated as follows:

- Smooth the Runt fluorescence field $\phi(x, y)$ with a Gaussian kernel $K_{\sigma_1}$ of width $\sigma_1 \approx 1$ cell diameter.
- Compute the smoothed gradient $\nabla(K_{\sigma_1} * \phi)$, and normalize its magnitude.
- Construct the local nematic tensor $\nabla(K_{\sigma_1} * \phi) \cdot \nabla(K_{\sigma_1} * \phi)^T$.
- Smooth the nematic tensor further by $\sigma_2 \approx \sigma_1$, so that the director is defined over the entire embryo scale (interpolating between stripes)
- The local Runt orientation is defined by the dominant eigenvector of the resulting tensor field.

## Additional data on PRG stripe, TLR stripe, and myosin orientation

In this section, we collect some additional data on the behavior of the genetic pattern of PRGs and TLRs as well as on the myosin orientation. *Appendix 1—figure 3* shows a quantification between the angle of the PRG Runt and the TLR Tartan, complementing *Figure 2A* in the main text. *Appendix 1—figure 4* shows that the PRG Eve is advected by tissue flow in the same way as Runt (see *Figure 2A*). *Appendix 1—figure 5* shows single-cell tracking data confirming the Lagrangian behavior of Runt.

Appendix 1—figures 6–8 show additional data on the local angular distribution of myosin and Runt (see also Appendix 1-figure 16). *Appendix 1—figure 6* complements *Figure 2F* by showing histograms of both the local myosin and the Runt orientation in the ventro-lateral region of the germband. In comparison to *Figure 2F*, where for simplicity all myosin junctions in the region under investigation were combined in a histogram, here, we proceeded in a two step fashion. The region was scanned with a $30\mu\text{m} \times 30\mu\text{m}$ window, for each window the mean orientation was calculated and subtracted from the angles of the junctions in the window, and the resulting distributions were joined. This process permits separating local spread from larger scale spatial gradients. *Appendix 1—figure 6* also shows the corresponding distributions for the Runt stripe orientations in the same region. Not only is the spread much smaller, the distribution also changes much less. The total spread of the Runt stripe distribution depends on the smoothing parameters used to compute the Runt gradient, but the dynamics of the distribution does not. *Appendix 1—figure 7* shows the spatial pattern of the local Runt angular distribution across the germband. Note the strong dissimilarity with the corresponding figure for myosin, *Appendix 1—figure 16*. Finally, *Appendix 1—figure 8* carries out a embryo-scale comparison of the local Runt and myosin distributions, as illustrated in *Appendix 1—figure 6*, using the entire ensemble of 10 embryos available to us. We compute the Kolmogorov–Smirnov, a measure of distance between two probability distributions (the maximal area between the two probability densities), of the local angular distributions across the germband, finding high values strongly indicative of disagreement between the two. Each window typically contains more than 100 junctions.

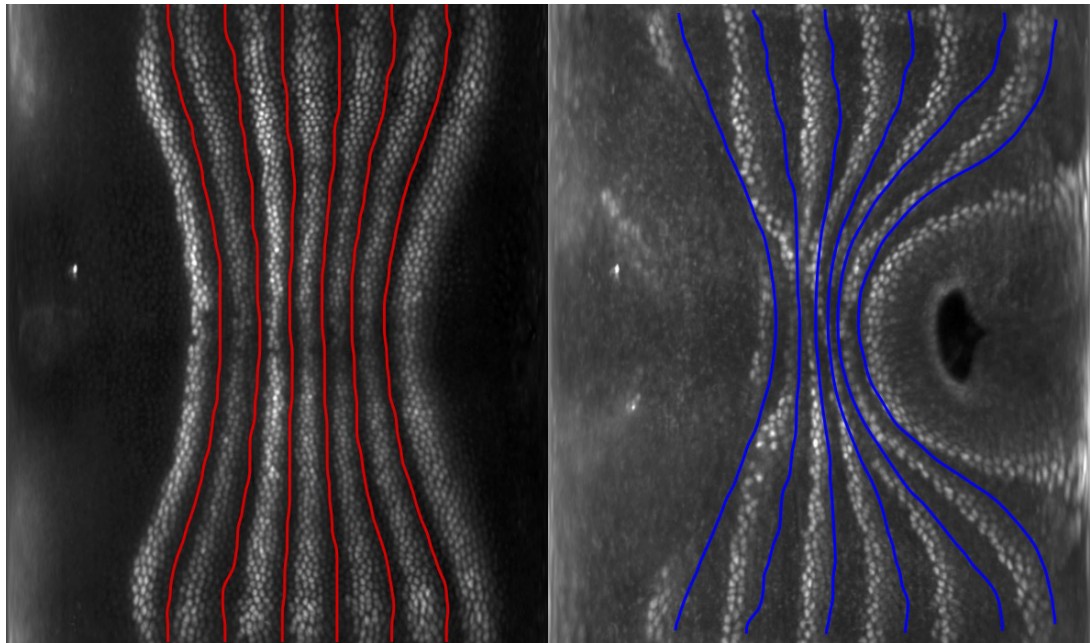

**Appendix 1—figure 4.** Eve stripes are advected by tissue flow. Eve stripes 5 min before ventral furrow (VF) initiation (left), and Eve stripes 20 min post VF initiation, with predicted inter-stripe locations based on advection (right).

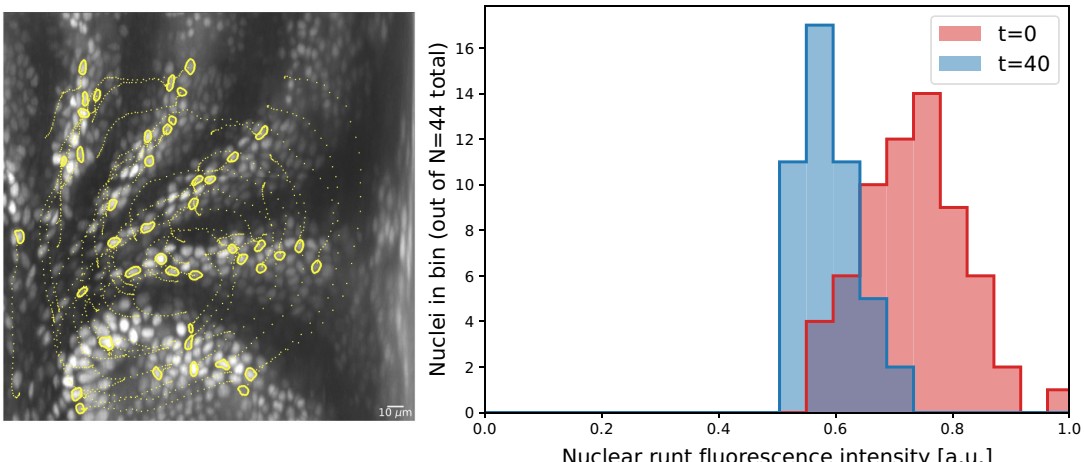

**Appendix 1—figure 5.** Nuclei initially expressing Runt expression after 20 min. Cell tracks obtained by semi-automatic cell tracking in Ilastik.

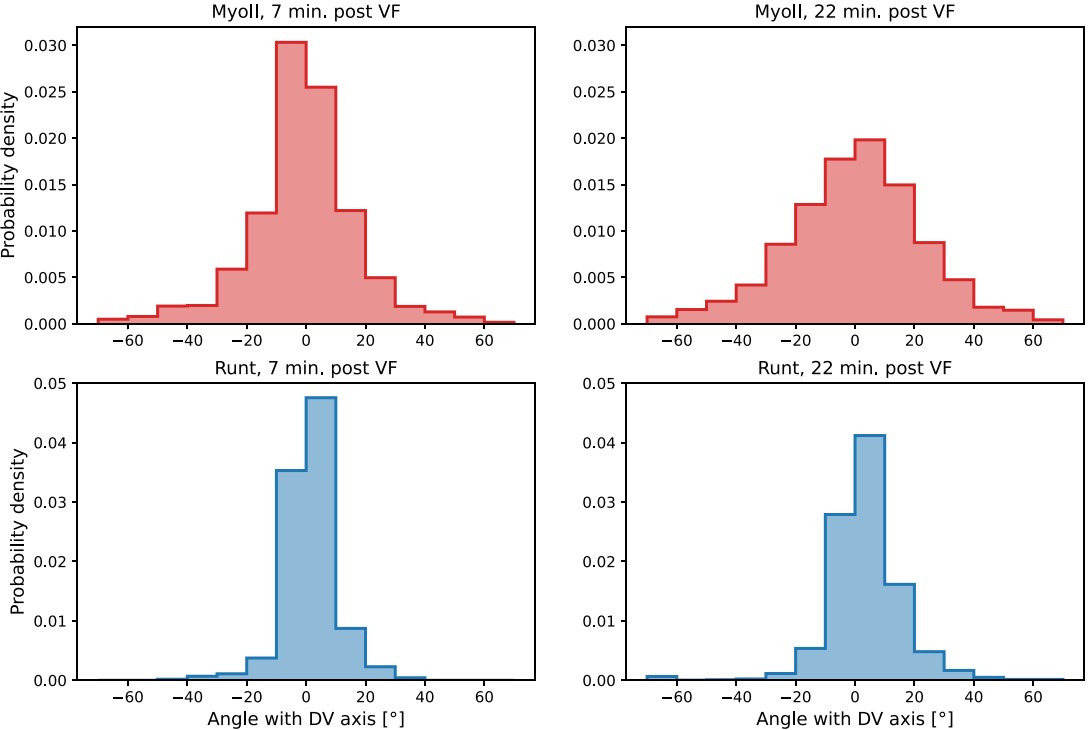

**Appendix 1—figure 6.** Local distribution of Runt and myosin orientations in the ventro-lateral region of two representative embryos 22 min post ventral furrow (VF) initiation. Runt angle and myosin data taken from to the regions shown *Figure 2E*. Window size used for querying the local distribution is $30\mu\text{m} \times 30\mu\text{m}$.

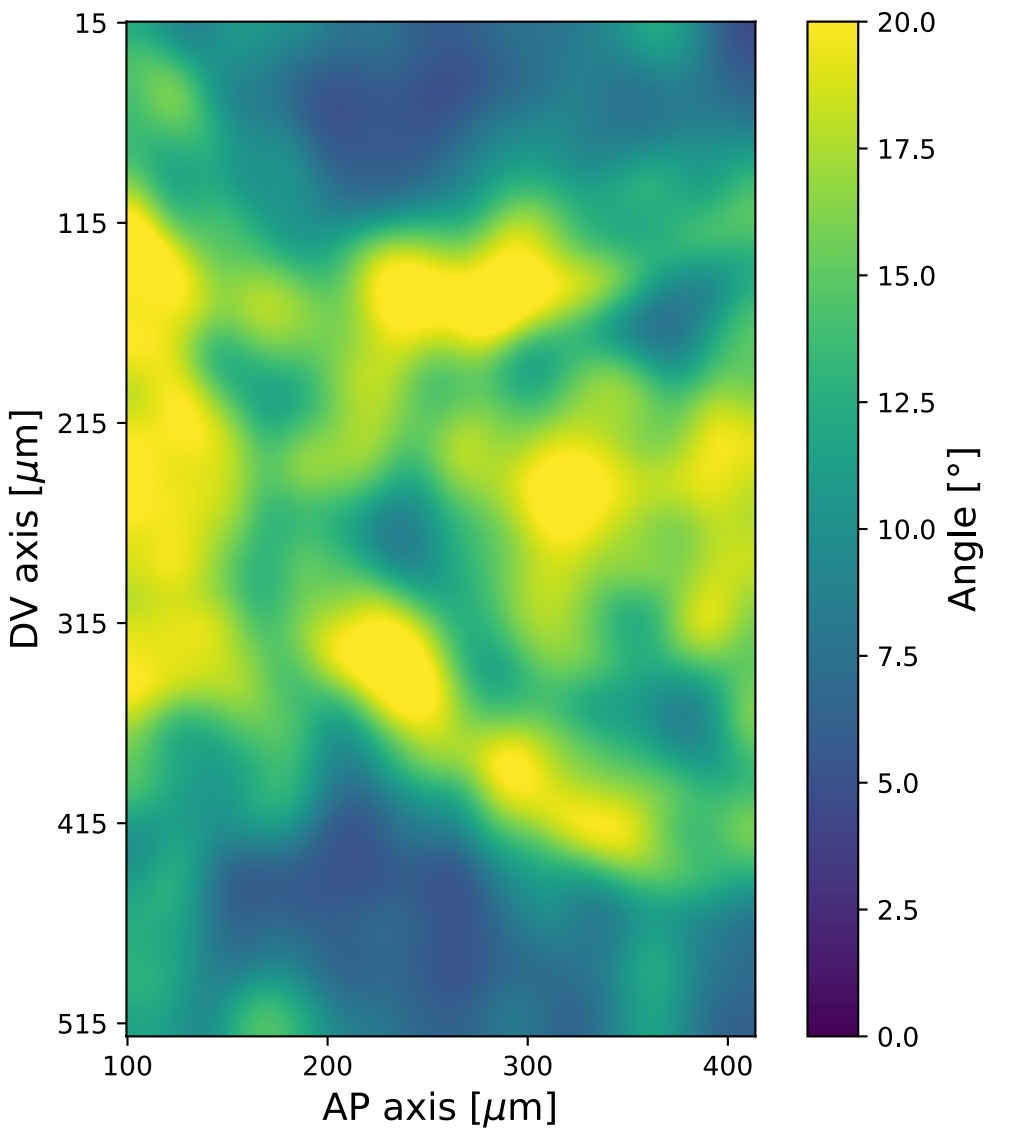

**Appendix 1—figure 7.** Spatial pattern of local standard deviation of Runt orientation, 20 min post ventral furrow (VF) initiation. Data from $N = 5$ Runt::LlamaTag-GFP embryos. Window size used for querying the local distribution is $30\mu\mathrm{m} \times 30\mu\mathrm{m}$.

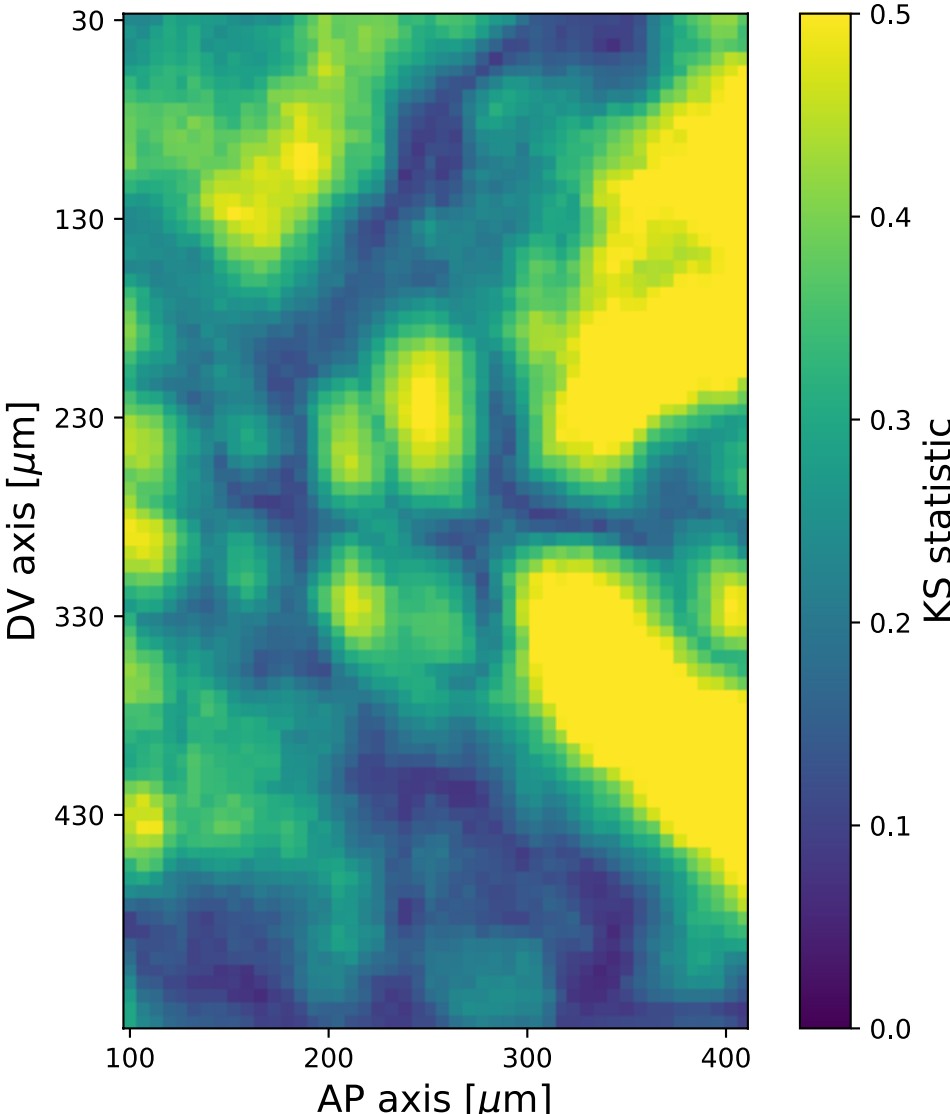

**Appendix 1—figure 8.** Kolmogorov–Smirnov test statistic comparing local angular distributions of Runt and myosin 20 min post ventral furrow (VF) initiation. Observed values of the test statistic strongly indicate that the two distributions are dissimilar. Runt angle from $N = 5$ Runt::LLamaTag-GFP embryos, myosin data from $N = 5$ Myosin::GFP embryos. Each local window contains ~100 junctions.

*Appendix 1—figure 9* completes the argument of *Figure 3* by showing the time course of PRG and myosin orientation as well as our prediction for the myosin orientation in the regions of all seven Runt stripes.

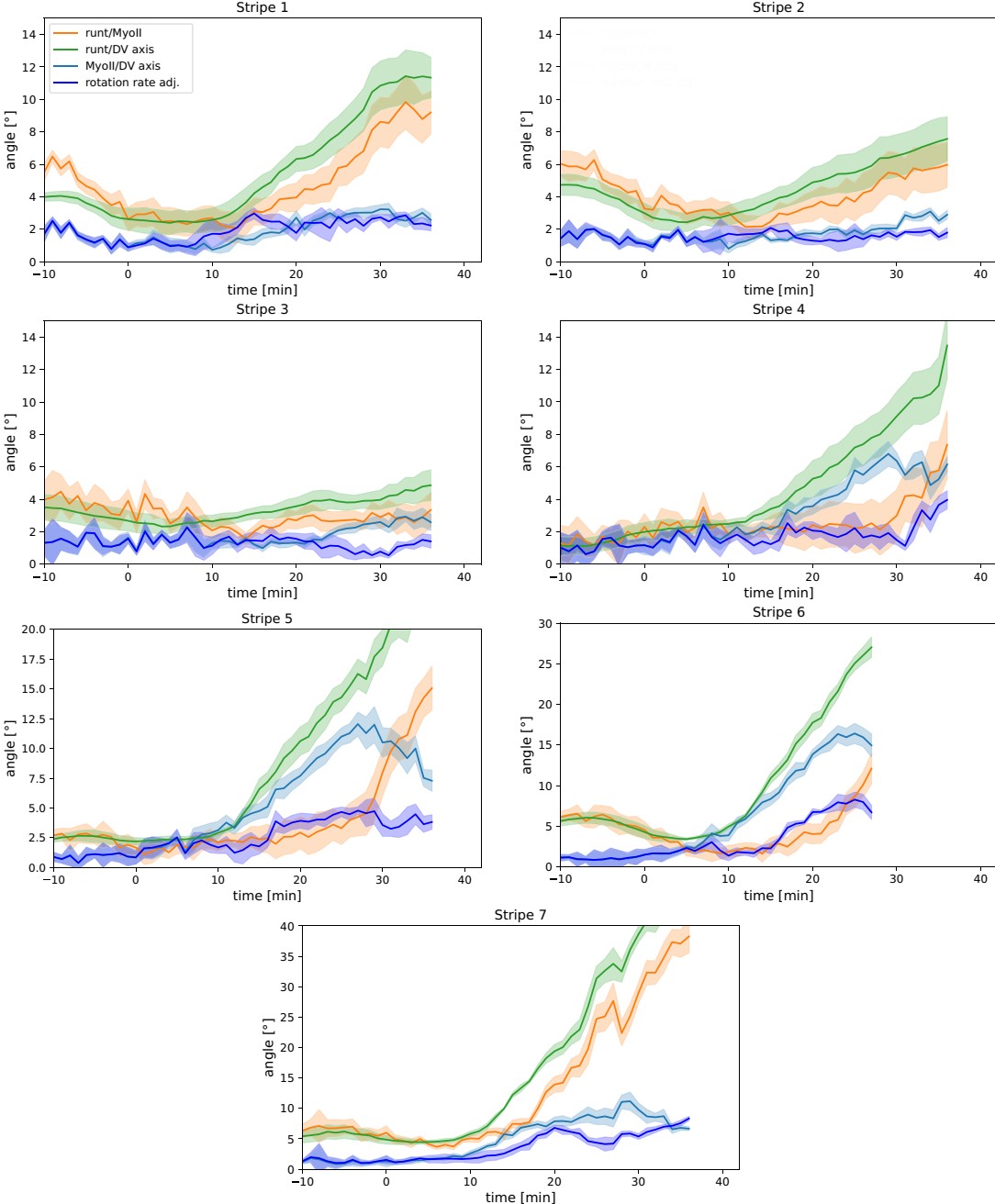

**Appendix 1—figure 9.** Per-Runt stripe averages of myosin and Runt orientations. Angle between myosin anisotropy orientation and Runt stripe, angle between Runt stripe and dorsal-ventral (DV) axis, angle between myosin anisotropy and DV axis and rotation rate-adjusted myosin/DV axis angle, averaged over the regions corresponding to Runt stripes 1–7. Data from $N = 5$ WT Myosin::GFP and $N = 5$ Runt::LlamaTag-GFP embryos. Data ends early in stripe 6 since the region becomes difficult to separate from stripe 7.

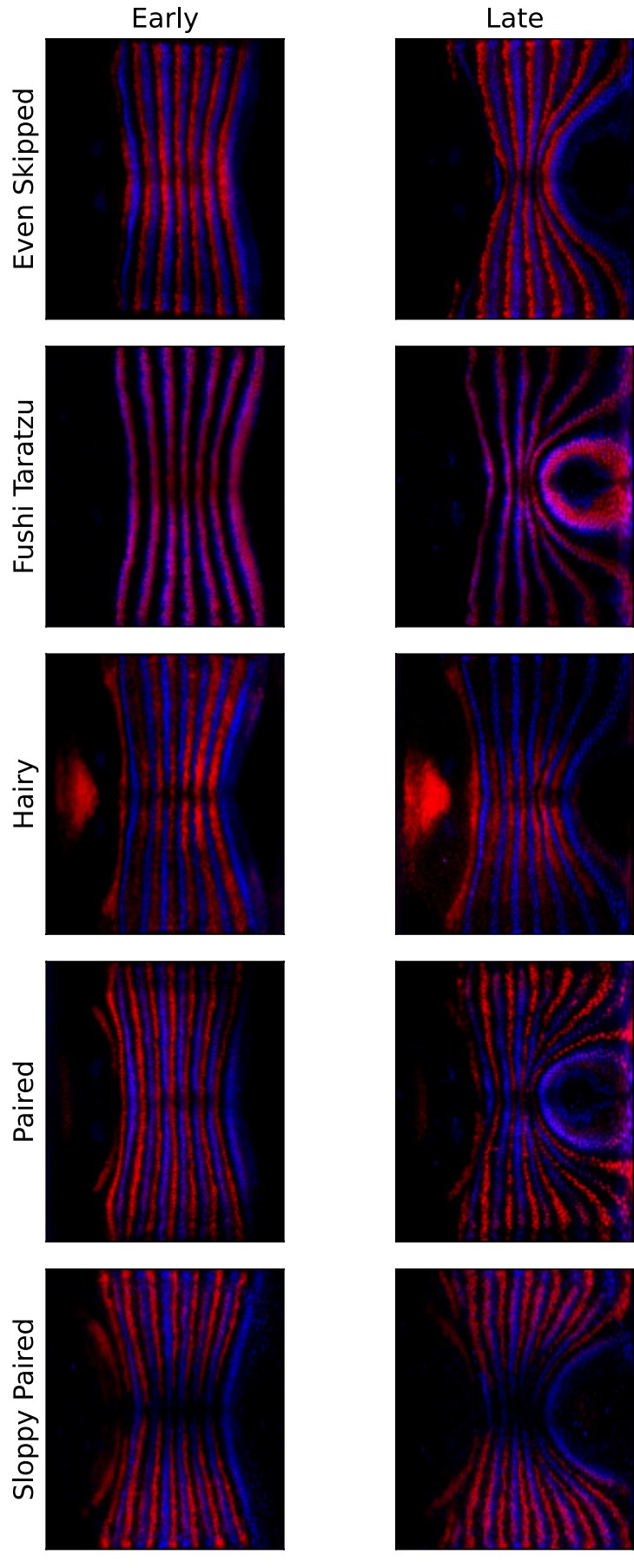

**Appendix 1—figure 10.** Pair rule genes (PRGs) remain parallel with respect to one another during germband extension (GBE). Pair-rule gene co-stains, showing Runt (blue) and another PRG (red, PRG indicated on the left), before the onset of GBE flow, and after significant flow has occurred (ca. 20 min later). The PRG stripes remain parallel to one another and do not penetrate one another, showing that Runt and the other PRGs are advected by tissue flow in the same way.

## Normalization of FRAP data

To obtain the FRAP curves shown in *Appendix 1—figure 3* from the raw confocal images, we used a two-step normalization procedure (*Wachsmuth, 2014*). The average raw intensity in the region of interest (ROI) containing the bleached junction will be denoted $I_t$. $t$ denotes time, with $t = 0$ being the first frame after bleaching. The bleached region is monitored for a total time $T$. In addition to the ROI containing the bleached junction, we also monitor three additional $5\mu\text{m} \times 5\mu\text{m}$ control regions that are neither bleached nor contain any junctions. Their average defines the background intensity $I_{\text{bg},t}$.

In a first step, we normalize the ROI intensity by the background, which also corrects for photobleaching, defining $\tilde{I}_t = I_t/I_{\text{bg},t}$. Next, the pre-bleach intensity $\tilde{I}_{\text{pre}}$ is the average normalized intensity in the ROI before bleaching. We define the FRAP signal as

$$Y_t = \frac{\tilde{I}_t}{\tilde{I}_{\text{pre}}}. \tag{9}$$

Note that the FRAP signal at $Y_{t=0}$ at the first frame recorded after bleaching is not zero. This is for two reasons: incomplete bleaching and the inevitable delay between the first frame and the end of bleaching. During this 1.5 s delay, non-bleached cytosolic myosin diffuses into the bleached region. Based on the FRAP curve $Y_t$, the myosin signal can be divided into three fractions:

- The fully mobile/incompletely bleached fraction $Y_0$.
- The slowly mobile fraction $Y_T - Y_0$ (signal recovered by the end of observation).
- The immobile fraction $1 - Y_T$ (signal not recovered by the end of observation).

Previous work *Munjal et al., 2015* used a different definition of the FRAP signal, namely

$$\hat{Y}_t = \frac{\tilde{I}_t - \tilde{I}_0}{\tilde{I}_{\text{pre}}}. \tag{10}$$

This definition leads to systematic underestimation of fluorescent recovery and overestimation of the immobile fraction, as can be seen from an example. Consider 90% effective photo-bleaching so that 10% of the pre-bleach signal is still present at the first time point post-bleach. To obtain a complete recovery using the $\hat{Y}_t$-measure would therefore require a 10% increase in fluorescent intensity over pre-bleach levels.

## FRAP data in mutant genotypes

We carried out FRAP measurements (*Wachsmuth, 2014*) as described above in two of the mutant genotypes described in the main text, $eve^{R13}$ and *Fat2-RNAi*.

In $eve^{R13}$, myosin dynamics deviates from WT in several respects. First, we observe increased variability in FRAP recovery across junctions. Because of this high variability, we are not able to resolve features in the FRAP curve later than 3 min. post bleaching, limiting the time over which we can compare it to WT. Second, we carried out two subsequent FRAP experiments for each embryo. When we stratify FRAPed junctions according to whether they were imaged first or second, we see that myosin recovery is incomplete in 'late' embryos, potentially because myosin recruitment has ceased (*Appendix 1—figure 12*). This is consistent with the slowdown of flow in $eve^{R13}$ after the onset of tissue flow in *Figure 4B*. We therefore only include the 'early' experiments in *Appendix 1—figure 11*. Finally, in the $eve^{R13}$ FRAP experiments, we achieved more complete bleaching (down to ~35% vs. 55% in WT). The $eve^{R13}$ and WT experiments were carried out 1 year apart, during which our experimental facilities have improved. When comparing the recovery kinetics, we are however interested in the ratios of fluorescent intensities at different timepoints (which then yields the recovery rate), so that the bleaching completeness drops out of the final result.

With all these caveats in mind, *Appendix 1—figure 11* shows the FRAP curve obtained for $eve^{R13}$ embryos. We observe that within 2 min post bleach, myosin intensity has recovered from 35% of pre-

bleach intensity to 90%. This corresponds to a recovery rate of ~2.4 min$^{-1}$ (interval corresponding to a 50% increase in fit error vs. the minimum: $1.95 - 2.9$ min$^{-1}$). To obtain the recovery rate $r$, we model the FRAP curve by a single exponential:

$$\text{FRAP}(t) = 1 - (1 - \text{Post-bleach intensity}) \times e^{-\log(r)t} \tag{11}$$

where $t$ is the time in minutes. A direct comparison to the WT FRAP curve is however problematic since the WT FRAP curve does not appear to follow a simple exponential recovery (**Appendix 1—figure 13**). Instead, the WT recovery appears biphasic, with a very rapid initial and a slower final phase, as described by the following equation:

$$\text{FRAP}(t) = 1 - (c_0 - \text{Post-bleach intensity}) \times e^{-\log(r_0)t} + (1 - c_0) \times e^{-\log(r_1)t} \tag{12}$$

Here, $c_0$ and $c_1$ parameterize the initial fast recovery. The second, slow recovery occurs with a rate of ~1.4 min$^{-1}$, and therefore slower than in $eve^{R13}$. Since junction rotation happens at a rate of ~1°/min, it is the slower recovery rate that is relevant for the myosin orientation behavior. Along the same vein, if we fit the single-exponential model to the WT data, ignoring the initial fast recovery for the purpose of the fit, we obtain a recovery rate of 1.9 min$^{-1}$. This suggests that the data is compatible with somewhat faster myosin kinetics in $eve^{R13}$ than in WT. However, we caution that the necessary difference modelling of the FRAP curve between the two genotypes renders the recovery rates hard to compare directly.

Finally, we also recorded FRAP curves in *Fat2-RNAi* mutants, which we present in **Appendix 1—figure 14**. Here, the FRAP curve resembles the biphasic behavior seen in the WT, with rapid early recovery. This is in accordance with the fact that the anterior-posterior patterning in these mutants remains intact (**Chanet et al., 2017**). The measurement error in *Fat2-RNAi* mutants was higher than in WT since the embryo only carried a fluorescent myosin marker, and not also a membrane marker.

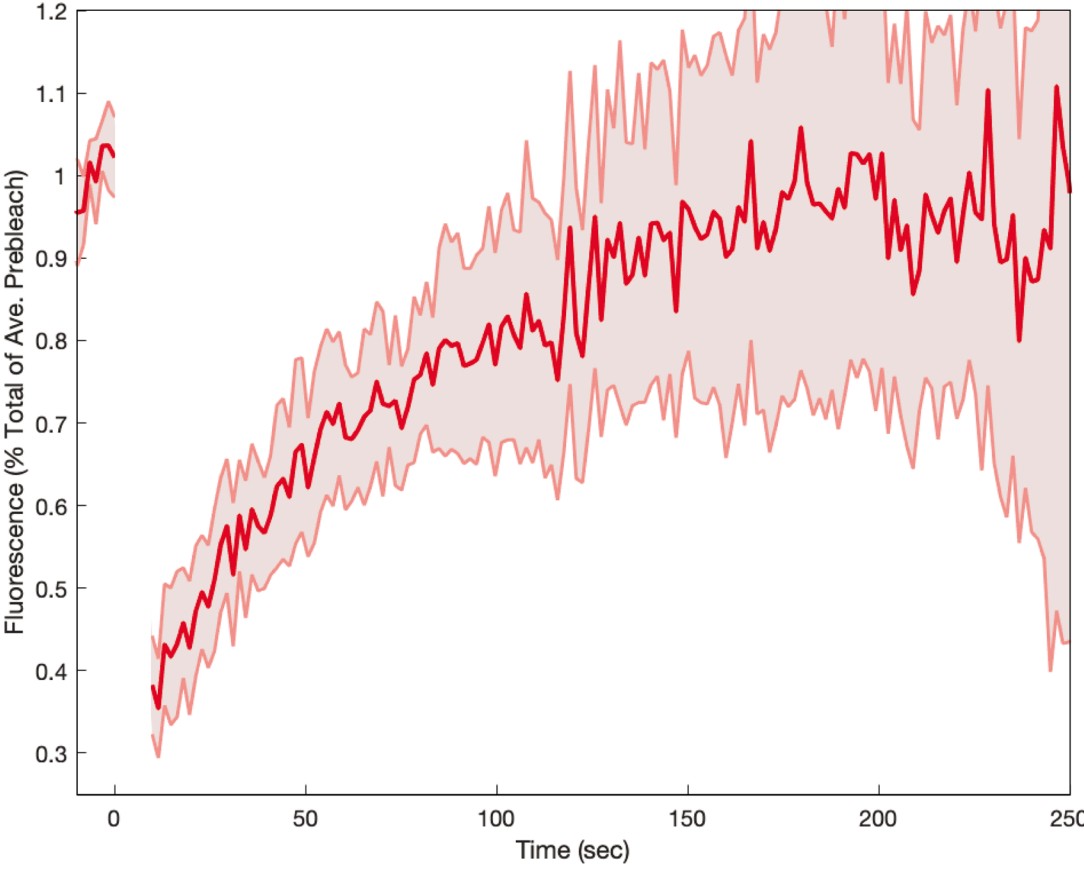

**Appendix 1—figure 11.** Myosin fluorescence recovery after photobleaching (FRAP) recovery in $eve^{R13}$ embryos. The data is compatible with the hypothesis that myosin kinetics is more rapid in $eve^{R13}$ embryos than in WT. Shaded error represent the standard error on the mean. $N = 5$ FRAPed junctions.

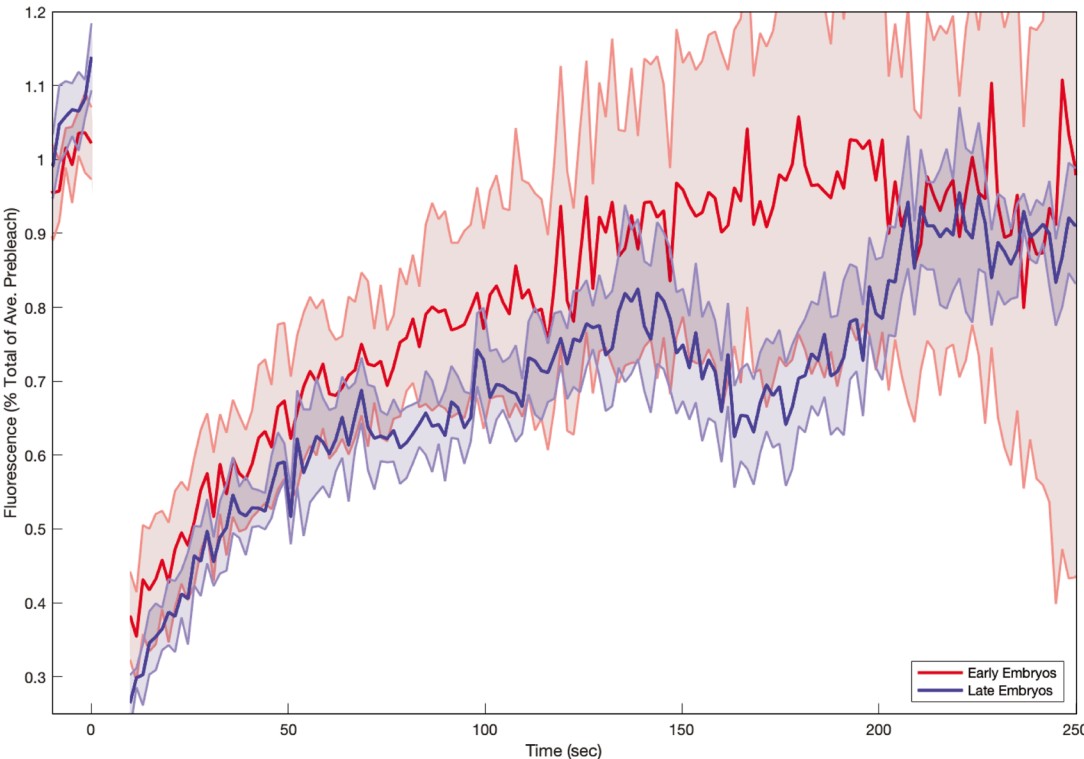

**Appendix 1—figure 12.** Myosin fluorescence recovery after photobleaching (FRAP) recovery in $eve^{R13}$ embryos as a function of embryo age. In $eve^{R13}$, moysin kinetics appears to depend sensitively on time, with less recovery in embryos during later germband extension (GBE). Shaded error represents the standard error on the mean. $N = 5$ (red) resp. $N = 6$ (blue) FRAPed junctions.

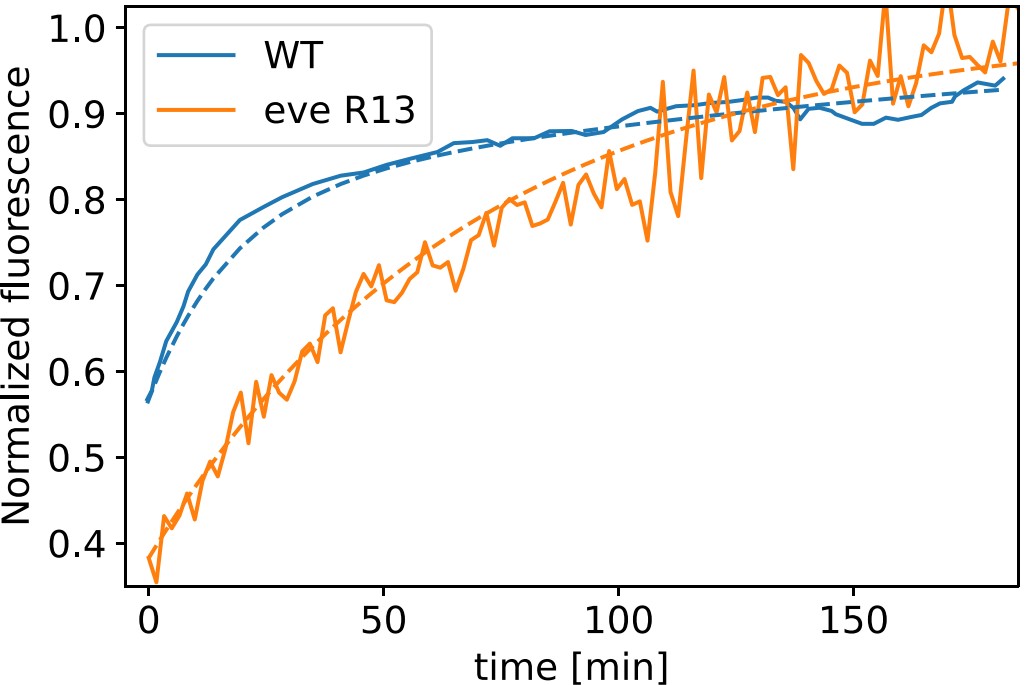

**Appendix 1—figure 13.** Myosin fluorescence recovery after photobleaching (FRAP) recovery in $eve^{R13}$ embryos, and WT, together with modeling fit. The $eve^{R13}$ curve is fit by a single exponential, whereas for the WT curve, two exponential terms are required.

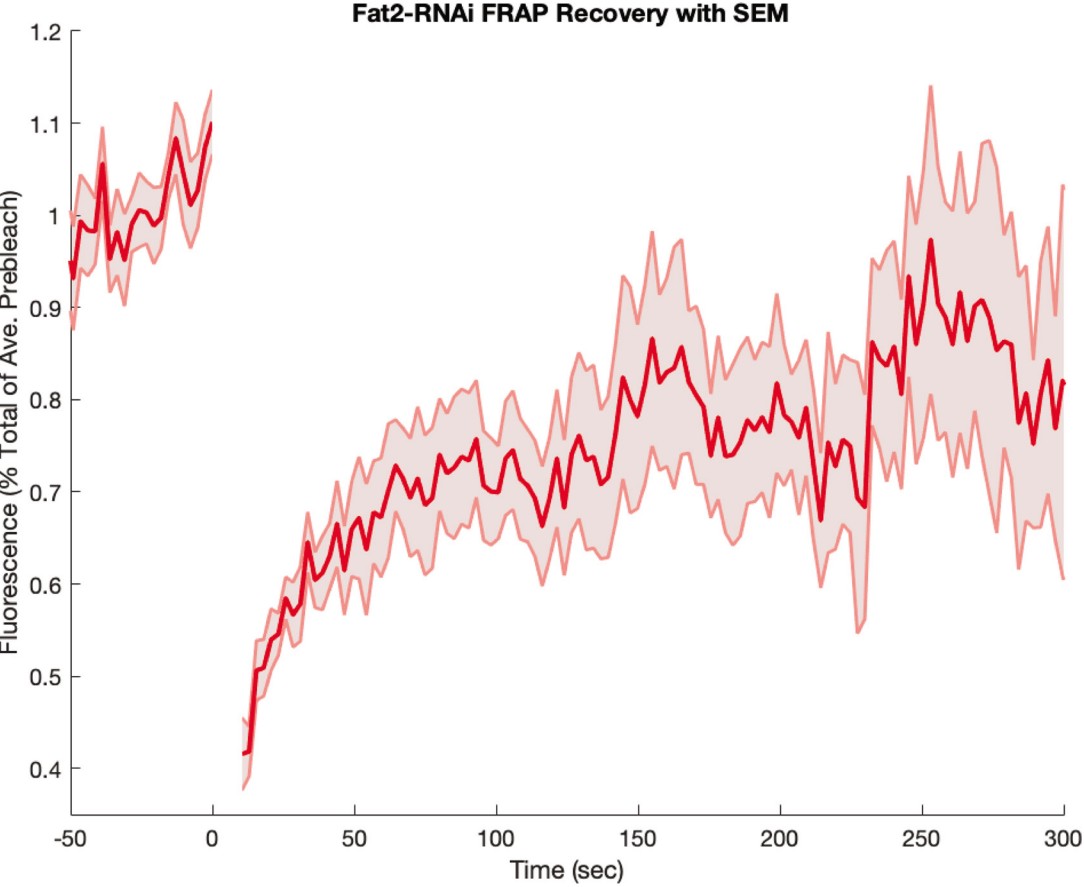

**Appendix 1—figure 14.** Myosin fluorescence recovery after photobleaching (FRAP) recovery in embryos from *Fat2-RNAi* mothers. Embryos were visualized with a fluorescent myosin only (and not also with a fluorescent membrane marker). Shaded error represents the standard error on the mean. $N = 8$ FRAPed junctions.

## Convective derivative analysis

Here, we present some additional data on the advection behavior of Runt stripes and myosin orientation. Instead of plotting the accumulated angle between Runt stripes/the myosin orientation and the DV axis, we can also analyze their rates of change. In *Appendix 1—figure 15* ,we compare the convective derivative, which measures the rate of change in the Lagrangian reference frame (flowing with the tissue), and the partial time derivative, which measures the rate of change in a fixed reference frame. This comparison shows that the Runt changes much less when viewed in the Lagrangian frame of reference, whereas for the myosin orientation the opposite is true.

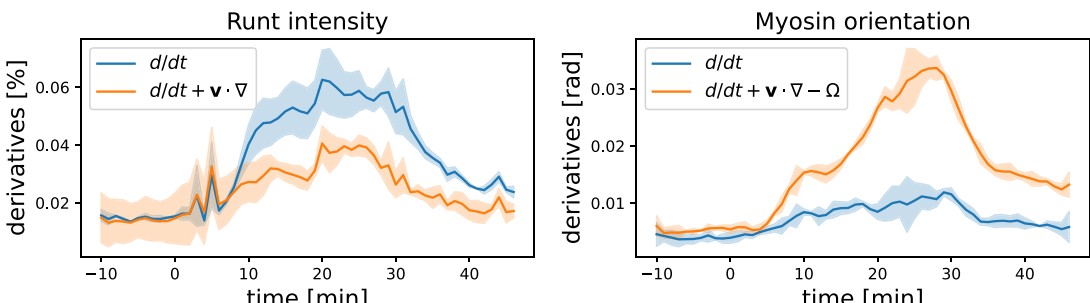

**Appendix 1—figure 15.** Convective vs. partial time derivatives of the Runt and myosin patterns. For the Runt intensity, the convective derivative is significantly lower than the partial derivative, while for the myosin orientation the opposite is the case.

## PRG gradient regression

In this section, we give details on the model used to test for correlations between the tissue-scale myosin pattern and the PRGs in *Appendix 1—figure 1*. As explained in the main text, previous work suggests that myosin is recruited to junctions between cells with differing levels of PRG expression. These cell-cell differences can be measured across the entire embryo by computing the gradient of the measured fluorescent intensity. The gradients are used as inputs in a linear regression model with the goal of predicting the myosin pattern.

Here, we give a step-by-step description of this process. We enumerate the different PRGs (e.g., Eve, Runt, and Ftz) with an index $i = 1, 2, \ldots$. The level of expression of a particular gene at a position $\mathbf{x}$ on the embryo surface is a scalar function $\phi_i(\mathbf{x})$. Before computing the gradients, the functions $\phi_i$ are convolved with a Gaussian kernel $K_{\sigma_1}$ with standard deviation $\sigma_1 \approx 1$ cell diameter. This ensures that when computing the gradient, we measure differences between cells and not spurious differences between the levels within and without a cell's nucleus. We denote the result $K_{\sigma_1} * \phi_i$. Next, we compute the magnitude of the gradient of the PRG fields, $|\nabla(K_{\sigma_1} * \phi_i)|^2$. Since we consider the germband away from the posterior pole, where the embryo is effectively cylindrical, $\nabla$ is computed using ordinary partial derivatives. The gradient magnitude field is then smoothed again, with a second kernel of standard deviation $\sigma_2 \approx 4$ cell diameters. This is done for two reasons: First, we are interested in predicting the embryo-scale myosin distribution. Second, smoothing over small scales will reduce errors, in particular those due to imperfect alignment of data from different embryos. Smoothing makes the model more generous, as can be seen from the limit case where both the myosin and the PRG patterns are completely smoothed out - the linear regression would report perfect correlation.

The linear regression model for the myosin magnitude takes the following form:

$$\lambda_{\text{regression}} = \sum_i \alpha_i K_{\sigma_2} * |\nabla(K_{\sigma_1} * \phi_i)| \tag{13}$$

Myosin magnitude is modeled by a linear superposition of PRG gradients with weights, each gene acting independently. The weights $\alpha_i$ represent the strength of each gene and are chosen by least-squares minimization. Note that the both the background levels of the PRG fluorescent signal and their image contrast are irrelevant since the former produces 0 gradient and the latter is absorbed by the fit weights. The model can be extended to allow cross terms:

$$\sum_{ij} \beta_{ij} \sqrt{(K_{\sigma_2} * |\nabla(K_{\sigma_1} * \phi_i)|) \cdot (K_{\sigma_2} * |\nabla(K_{\sigma_1} * \phi_j)|)}. \tag{14}$$

Cross terms model recruitment of myosin specifically to edges between two different PRG gene stripes. In our model, we consider three PRGs, Eve, Runt, and Ftz and allow for both linear terms and an Eve/Runt cross term.

Both the myosin and the PRG patterns depend on time $t$. This raises a potential complication since the myosin pattern is would be expected to react to the PRGs with a delay whose duration is unknown. We carried out the linear regression to fit the myosin pattern at time $t_{\text{myosin}} = 10$ min post VF formation when the characteristic GBE myosin pattern has already been established, but no significant deformation due to tissue flow has yet occurred. The PRG gradients are evaluated at an earlier time $t_{\text{PRG}} < t_{\text{myosin}}$ that we can vary as an additional fit parameter. However, the PRG pattern changes little in the 20 min preceding $t_{\text{myosin}}$ and we do not find a good agreement between the PRG-based model and the myosin distribution for any value of $t_{\text{PRG}}$.

### Nonlinear PRG-based model

The PRG regression model presented in the main text, *Equation 1*, is the simplest possible relation between PRGs and myosin anisotropy. As we show, it fails to provide a good account of the observed myosin pattern. But it is easy to imagine that the relationship between myosin recruitment and PRGs is more complicated, and in particular, nonlinear. In this section, we address this question using *nonlinear regression*. This means we generalize our model as follows:

$$\text{myosin} = \sum_{\text{PRG } i} f_i(\text{all PRGs, all PRG gradients}) \times \text{gradient of PRG } i \tag{15}$$

We replace the constant weighting factor for the gradient of each PRG by a nonlinear function that can depend on the expression levels and gradients of all PRGs. This function is assumed to be a polynomial of degree $d$. Degree $d = 0$ represents the original model with constant weighting factors. The choice of model above implies that every term in the nonlinear regression model contains at least one PRG-gradient factor, reflecting the fact that myosin anisotropy cannot be generated by a scalar gene expression pattern alone. The coefficients of the polynomials $f_i$ are optimized via linear regression to fit the observed myosin pattern. PRG gradients and smoothed PRG expression patterns are computed as above, and all model inputs are rescaled to range $[0, 1]$ before fitting the model. We consider the PRGs Runt, Eve, Ftz, Paired, Sloppy-Paired, as well as all of their gradients.

To evaluate the model, we need to take some additional care compared to the simple linear model of *Equation 19*. Already at polynomial degree $d = 2$, the nonlinear regression model has a very large number of parameters, 330. This means that it is to be expected that almost *any* pattern could be fit by *Equation 14*. Successful fitting would therefore not constitute strong evidence for a relationship between PRGs and myosin.

Indeed, Petkova and coworkers *Petkova et al., 2019* have found that the position of a cell along the AP axis can be decoded from the joint gap gene expression pattern (immediately upstream of the PRGs) with high fidelity. In agreement with this, we find that with $d = 2$, we can fit the AP and DV coordinates of the embryo with low error (mean absolute error of > 5%) using *Equation 14*. Once we know that these coordinate values can be fit from PRGs, it is clear that almost any spatial pattern could be obtained.

This issue – an expressive model that can fit almost any pattern – is well known in the machine learning and statistics literature. The solution is to use separate training and test sets. This means that the dataset is split into two disjoint sets. The training set is used to optimize the model parameters, and the test set, to test how well the model obtained performs when predicting data it has not yet 'seen,' and thereby faithfully evaluate the performance of models with large numbers of parameters. For this to work, the training and test set have to be independent. In the context of the *Drosophila* embryo, we implement this by splitting the germ band into two parts along the AP-axis, using one half to fit the model *Equation 14*, and the other to evaluate its quality. The rationale is that if the PRGs contain the positional information instructing the myosin pattern then the same model $\mathrm{PRGs} \mapsto \mathrm{myosin}$ should work all over the germ band. We proceeded this way because other options for generating splits do not yield 'independent' test/train sets. For example, the fact that myosin/ PRG patterns are strongly stereotyped patterns means that using samples from different embryos is not a good option, and the fact that we study smoothly varying patterns means that randomly splitting coordinate positions (i.e., image pixels) does not work either, because neighboring pixels are strongly correlated. We tried different choices of the AP split (anterior vs. posterior, posterior vs. anterior, middle vs. anterior and posterior, etc.), with very similar results. The numbers quoted below are for training on the posterior part of the germ band (57–87% of embryo length), and testing on the anterior part (27–57% – the total region under investigation is the expression domain of PRGs before GBE). Errors quoted below are mean absolute errors, normalized by the mean of the regression target.

By directly optimizing the model coefficients for $d = 2$, we find that the model can fit the training set almost perfectly (error below 5%) and produces a pattern visually extremely similar to the observed one. However, on the test set, the test error was extremely large, at 150%. This suggests that the model did not successfully capture a general relationship between PRGs and myosin.

A common method to improve the gap between training and testing performance is to use so-called *regularization*. Regularization is a way to penalize more complex models during the fitting procedure: the optimization procedure now simultaneously minimizes the model error and a measure of the model complexity. The two most widely used and successful regularization techniques for linear models are L1 and L2 regularization (the regularization parameters were optimized by cross-validation). We found that an L2-regularized model still fit the training data well (training error below 10%), but still suffered serious test error (>60%). The best results were obtained with L1-regularization, which seeks to minimize the number of variables included in the model. Here, train and test errors were roughly similar, but still large, at 40%. For both regularized models, allowing more non-linear models (degree $d > 2$) did not improve performance.

Crucially, the best-performing L1-regularization only contained a two nonzero terms, one of which accounted for almost 80% in the model output variance. This term corresponded to the product of

Sloppy-Paired expression intensity and the gradient Fushi-Taratzu expression intensity. Given that these PRGs are know to have a much weaker influence on GBE than, for example, Eve or Runt (*Irvine and Wieschaus, 1994*), it is very likely that the L1-model only exploited an accidental correlation. Further, the L1-model predicted myosin field displayed a strongly striped pattern, reflecting the stripes of PRG Paired, whereas the actual myosin pattern shows no such stripes at all.

Finally, the type of nonlinear regression model we considered above is of course not the only possibility. We also experimented with so-called symbolic regression, a technique based on genetic programming, as implemented in the Python package gplearn. However, we found the genetic algorithm difficult to control and quite slow, while not delivering better results than nonlinear regression. In particular, we used the following test case: fit the known myosin pattern, given the spatial coordinates $x, y$. With polynomial nonlinear regression, this task is easily accomplished with ease within milliseconds, whereas it took significant parameter tuning and several minutes of training with gplearn. Further, the output program was difficult to interpret, in contrast to L1-regularized regression.

Overall, we conclude that even more complicated, nonlinear models do not account well for the relationship between PRGs and myosin.

## Static source model

In this section, we supply a quantitative model describing the behavior of junctional myosin under the hypothesis of recruitment by a static source, supporting the argument made in section 'Results.' During tissue flow with velocity $\mathbf{v}$, cell junctions rotate with a rate equal to one-half of the vorticity $\omega = \nabla \times \mathbf{v}$ (*Landau and Lifshitz, 1987*). Note that the tissue shear does not significantly reorient cell junctions: since GBE proceeds primarily by cell intercalation, cell shapes are not changed by the tissue-level shear. Further, junction lengthening and shrinkage do not affect junction orientation. We weight junctions by their average myosin intensity, i.e., total myosin signal divided by junction length, so that junction length does not affect the weighting.

We assume that myosin is recruited to a static source whose orientation of the source is defined geometrically: it is parallel to the direction of maximal curvature of the embryo surface. This defines the DV axis for the purpose of the measurements reported in *Appendix 1—figures 2–4*. Myosin is assumed to be recruited to edges aligned with the source and to detach from all edges with a constant rate $1/\tau$. These assumptions can be encoded into equations in different, equivalent ways. In main text *Equation 2*, we considered a single junction, below we consider the local myosin nematic tensor constructed from detected MRJ in a small tissue patch, and in Sect. I.7.1 we consider the entire angular distribution of myosin.

This yields the following equations for the local myosin tensor $Q_m$ as defined in Sect. I.1.4:

$$\partial_t Q_m + [\Omega, Q_m] = -Q_m/\tau + \Gamma \tag{16}$$

Here, $\Omega$ is the vorticity matrix $\Omega_{ij} = (\partial_i v_j - \partial_j v_i)/2$ and $\Gamma$ is the source tensor, parallel to the DV axis. Advection has been neglected (see Sect. I.7.2). In the steady state, this equation is solved by

$$Q_m = R_{\omega\tau/2}(\Gamma) \tag{17}$$

where $R_{\omega\tau/2}$ represents a rotation by angle $\omega\tau/2$. The steady state is a good approximation for most of GBE, since after flow onset, the velocity pattern flow is relatively steady compared to the myosin lifetime $\tau \approx 5\text{min}$.

Crucially, *Equation 16* shows that the dominant eigenvector of $Q_m$, i.e., the direction of myosin anisotropy, is given by the dominant eigenvector of $\Gamma$, rotated by an angle of $\omega\tau/2$. This prediction is completely independent of the magnitude of the source $\Gamma$!

## FRAP-measured myosin lifetime and effective lifetime

We note that the effective lifetime $\tau$ in the model *Equation 15* is not necessarily equal to the time $\tau_{\text{bound}}$ an individual myosin motor remains bound to a junction (as measured by FRAP, for example). This can be seen by considering an example scenario in which myosin motors detach extremely rapidly, but the myosin concentration on an edge is controlled by a long-lived actor up the regulatory chain setting, for example, the rate of myosin phosphorylation. Due to the possible persistence of actors upstream of junctional myosin, the effective lifetime $\tau$ is larger or equal to the bound time of

individual motors. This is a possible explanation for the discrepancy between the FRAP measured $\tau_{\text{bound}}$ and the inferred effective $\tau$ in **Figure 3**.

## Model for myosin angular distribution

We next show how to predict both the mean and the standard deviation of the distribution of orientations of myosin-carrying edges. To this end, we consider a simple model for the angular distribution $m(t, \theta)$ of myosin in a tissue patch. $\theta = 0$ is taken to be the local orientation of the static source. A simple model for the time evolution of this distribution, derived from the single-junction dynamics in main text **Equation 2**, reads as follows:

$$\partial_t m = -\frac{\omega}{2}\partial_\theta m - \frac{1}{\tau}m + \frac{1}{\tau}\Gamma \qquad (18)$$

The first term describes the rotation of edges by the vorticity $\omega$, the second the detachment of myosin after an effective lifetime $\tau$, and the third term $\Gamma/\tau$ is the static source term, peaked around $\theta = 0$. From this, one can derive **Equation 3** for the time evolution of the mean angle $\bar{\theta}$. **Equation 17** is the equation used to obtain the simulated histograms in **Appendix 1—figure 3**, with the choice $\Gamma = e^{-k sin^2(\theta)}$. In the steady state,

$$\partial_t m = 0 \Rightarrow m(\theta) = -\frac{\omega\tau}{2}\partial_\theta m + \Gamma. \qquad (19)$$

From this, the moments of $m$ can be found by partial integration. The 0th moment is independent of $\omega$, so that the overall amount of myosin on edges and the normalization of $m$ are not affected by vorticity. For the mean $\mu$ and variance $\sigma^2$, one finds

$$\mu - \mu_0 = \frac{\omega\tau}{2} \qquad (20)$$

$$\sigma^2 - \sigma_0^2 = \frac{(\omega\tau)^2}{4} \qquad (21)$$

Here, $_0$ denotes the values in the case of $\omega = 0$. Notably, the shifts in mean and variance are completely independent of the form of the source term $\Gamma$. This means that the predictions again depend on only one parameter, $\tau$.

**Equation 19** can be generalized to the case where the vorticity varies in time:

$$\mu = \int_0^t \frac{\omega(t')}{2}e^{(t-t')/\tau}dt' \qquad (22)$$

This is the equation that is used to generate the predicted myosin/DV axis angle shown in **Appendix 1—figures 3 and 4**. For the prediction of the spatial average myosin/DV axis angle, we add a constant of 1.5° to **Equation 21** to account for the nonzero myosin/DV axis angle before the onset of vorticity.

One can also take into account the fact that then angle $\theta$ is only defined module $\pi$ and consider moments of $e^{i2\theta}$ instead of $\theta$, yielding similar results, for example $\mu - \mu_0 = \tan^{-1}(\omega\tau)/2$.

## Prediction of variance of myosin angular distribution

Based on our model, we can predict both the mean myosin orientation as well as the variance of the angular distribution. In the main text, we presented simulated histograms (according to **Equation 17**), which show striking qualitative agreement of the predicted behavior and the observed broadening of the angular distribution once vorticity sets in. In the tissue patch shown in the main text (**Appendix 1—figure 2**), we find a change of the mean of $\mu_2 - \mu_1 = 16°$ and a change in variances of $\sqrt{\sigma_2^2 - \sigma_2^2} = 17°$ between the two timepoints analyzed, in line with the prediction $\mu_2 - \mu_1 = \sqrt{\sigma_2^2 - \sigma_2^2}$ of **Equation 20**. In this section, we check the prediction of **Equation 20** on the entire ensemble of $N = 5$ embryos and over the entire germband.

However, quantitatively, the behavior of the variance is more complicated than that of the mean angle. The variance depends on the strength of the anisotropic myosin recruitment compared to the isotropic background as well as on the myosin signal-to-noise ratio, which is not the case for the mean. If the strength of the anisotropic source or the overall levels of junctional myosin (and hence the

signal-to-noise ratio) decreases, as they do towards the end of GBE, the variance will increase, even in the absence of vorticity. On the other hand, a transient infcrease of the anisotropic recruitment leads to a decrease in variance, as is observed during the strong increase in myosin anisotropy due to strain generated by the ventral furrow during early GBE. Therefore, the vorticity effect *Equation 20* only represents one contribution to the variance. Further, the local variance depends on the size of the local tissue patch queried. If this size is chosen too large, tissue-scale gradients (e.g., of vorticity) contribute and increase the variance. If it is chosen too small, the ensemble calculation stops making sense.

Below we show both the temporal and spatial correlation of the measured local standard deviation (in 50 µm× 50 µm windows) and the vorticity-based prediction in the germband. We exclude a strip of 50 µm width around to the VF from the analysis since here the pulling effects of the VF dominate. We take the reference time for the initial variance at $t = 10$ min post VF initiation when the anisotropic myosin pattern is mostly established. The results of the prediction of *Equation 20* are shown in *Appendix 1—figure 16*. The most important aspects of the spatial and temporal behavior of the variance are explained by the vorticity model.

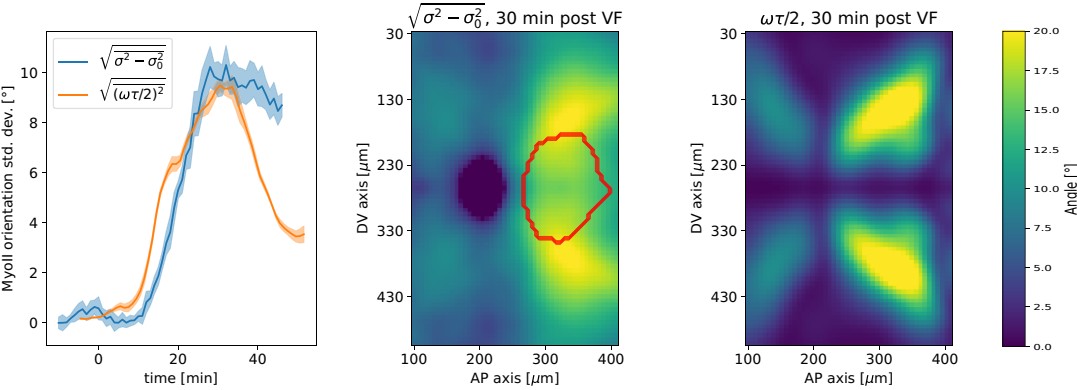

**Appendix 1—figure 16.** Embryo-scale prediction of myosin angular distribution width. From left to right: spatial average over the germband of standard deviation of myosin angular distribution and vorticity contribution to standard deviation – heatmap of change in standard deviation of myosin myosin angular distribution, showing germband only. Red outline indicates region of the invaginating posterior midgut – heatmap of vorticity contribution to myosin standard deviation.

## Additional effects in static source model

In this section, we discuss a number of effects not included in the model *Equations 15 and 17* and why we believe they are negligible

### Effect of advection

In addition to being rotated by flow, edges are also transported across the embryo surface by advection. This affects the observed spatial pattern of the myosin/DV axis angle. Advection can be accounted for by tracing the trajectory of a tissue patch back along the flow lines of the velocity field and correspond to adding a term $+\mathbf{v} \cdot \nabla m$ to the left hand side of *Equation 15*. However, we do not do so in our vorticity-based prediction of the myosin/DV axis angle. Advection is expected to have a small effect since (a) the orientation of the geometrically defined DV axis varies very little across the embryo surface, (b) the flow is not fast compared to the myosin lifetime $\tau$, and (c) around a vortex, the vorticity is approximately constant along flow lines. We can therefore neglect the effects of advection to first order and obtain a much simpler model wherein the myosin/DV axis angle at a given position is predicted from the vorticity at that same position. The residual effects of advection are one contribution to our model's error.

### Tissue rotation on curved surfaces

The fly embryo's surface is curved, and therefore, accounting for tissue rotation due to flow requires some mathematical care. To fix notation, we call the surface of interest $S$. Tissue flow defines a maps $\phi_t : S \mapsto S$, which send a point from its initial position at time 0 to its position on $S$ at time $t$.

The maps $\phi_t$ define trajectories (flow lines): $x(t) = \phi_t(x_0)$, where $x_0$ is a point on $S$. To see how tissue flow affects quantities of interest, in particular the myosin orientation, we need to *transport* them along trajectories. Myosin orientation is defined by a tensor $Q_m$, but since we are interested its the dominant eigenvector $\mathbf{n}$ only, we can think of myosin orientation as a vector field (In differential geometry terms, it seems actually to make more sense to think of myosin orientation as a 1-form, i.e., a map edge vector $\mapsto$ myosin concentration). Differential geometry offers different notions of "transport". Here, we need a mathematical answer to the question: What will the myosin orientation look like at time $t$ if myosin were passively advected, given the initial myosin orientation and the flow lines?

This answer is provided by the notion of pushforward and pullback, or Lie transport. When calculating the Lie transport of a vector field, it is rotated and sheared according to the deformation of the tissue patch around every vector, which is what we are looking for *Lee, 2013*. The tissue shear and rotation around a streamline $x(t)$ depends on the relative motion of nearby streamlines. Differential geometry also defines the notion of parallel transport, where the vector transformation is independent of nearby streamlines (*Lee, 2013*). Instead, it is determined by the local geometry; for example, a vector parallel transported along great circles of a sphere remains parallel to the great circles. Parallel transport requires defining a connection, which encodes the surfaces geometry. Very often, the Levi-Civita connection is used, which encodes how the surface $S$ is curved and is based on its Riemannian metric $g$. $g$ measures the lengths and angles of vector on the surface; for example, angles on a sphere behave differently than on a flat surface, with triangles having an angle sum of >180°. Crucially, Lie transport is completely independent of any connection or metric.

Lie transport can also be used to define a Lie derivative, measuring the infinitesimal change due to Lie transport. In flat space, it reduces to the usual convective derivative. Infinitesimally, the flow is defined by a velocity field $v^i$, and the Lie derivative of a (co)vector $w_j$ is given by *Lee, 2013*.

$$(\mathcal{L}_a w)_i = (v^j \partial_j) w_i + (\partial_i v^j) w_j \tag{23}$$

The second term represents the transformation due to tissue deformation. Note that in *Equation 22*, we can replace any partial derivatives $\partial_i$ by the covariant derivatives $\nabla_i$:

$$(\mathcal{L}_a w)_j = (v^i \nabla_i) w_j + (\nabla_i v^j) w_j \tag{24}$$

Here is where we use the metric $g$: because it allows us to measure angles and length, we can use it to decompose the transformation into a shear and a rotation, and measure the rotation angle. However, the vorticity, the antisymmetric part of the matrix $\nabla_i v^j$, turns out to be independent of the metric (see *Weinberg, 1972*, chapter 4.7–4.8), and is always given by the coordinate-based formula $\partial_x v_y - \partial_y v_x$. Therefore, the vorticity, which in our model determines how much the myosin orientation changes, is not actually affected by curvature effects.

Next, we use the vorticity to model the deviation of the myosin orientation away from a reference orientation, the direction of maximal curvature. If the direction of maximal curvature varied significantly across our cylindrical chart due to geometry variations, this calculation would be invalid. However, both WT and *Fat2-RNAi* eggs are fairly rotationally symmetric around the AP-axis (*Fat2-RNAi* even more than WT), and therefore the axis of maximum curvature everywhere points in the azimuthal direction in the cylindrical chart, with only very small deviations.

In any case, in the WT embryo, away from the poles, curvatures is very low: the average Gaussian curvature, excluding 10% of the embryo length around each pole, corresponds to a curvature radius of 185$\mu$m.

Curvature may however have an influence on tissue by inducing tissue deformation as formerly flat tissue moves into a region of high curvature and is bent and/or compressed. Since strain is not included in our model, and this effect depends on the 3D nature of the epithelial sheet, and is only relevant where curvature changes rapidly (i.e., only near the poles, even in a *Fat2-RNAi* embryo), it is beyond the scope of this work.

## Effect of myosin-feedback

If we assume that the source $\Gamma$ is a function of the tension on a junction, it is likely that $\Gamma$ itself could depend on the level $m$ of myosin on that junction. As long as this dependence is linear, it only renormalizes the value of $\tau$ and has no novel effect. Here, a graphical analysis is helpful in the $m$-$\dot{m}$ plane is helpful. Myosin detachment, i.e., $\dot{m} = -m/\tau$ represents a straight line in this plane, whose

intersections with $\dot{m} = \Gamma(m, \theta)$ defines the steady state of $m$ on an edge of a given orientation. In the case of an $m$-independent source, this is just a horizontal line. Even if $\Gamma$ depends on $m$, as long as there is only a single intersection between $\Gamma(m, \theta)$ and $m/\tau$, the dynamics is qualitatively unaltered. However, the lifetime of high-myosin edges will be enhanced. Two intersections signal runaway unstable behavior in which myosin levels on an edge ratcheted up without bounds, clearly contrary to observations. In the case of three intersections, there are two stable equilibrium values and high myosin levels can sustain themselves through positive feedback. Since now myosin levels need not decay if edges rotate out of alignment with the static source, one would not expect to observe the recovery effect shown in *Appendix 1—figure 3*.

## Effects of modification of myosin lifetime by static source

The equilibrium myosin concentration $\bar{m}$ on an edge is determined by the balance of the attachment and detachment rates $k_{\text{on}}$ and $k_{\text{off}}$ (equivalent to the lifetime and source in the previous section): $\bar{m} = k_{\text{on}}/k_{\text{off}}$. Therefore, it is possible to control the myosin distribution by either parameter, and a purported static source could influence either $k_{\text{on}}$ or $k_{\text{off}}$.

However, the rate by which the myosin concentration converges to the equilibrium value differs between the two scenarios. Consider an edge which rotates from an initial orientation $\theta_1$ with equilibrium value $\bar{m}_1$ to an orientation $\theta_2$ with equilibrium value $\bar{m}_2 = \bar{m}_1/2$. This can either happen if $k_{\text{on}, 1}$ decreases by a factor of 2, or if $k_{\text{off}, 1}$ increases by a factor of 2. In the the $k_{\text{on}}$-case, the edge maintains an elevated myosin level for a time $1/k_{\text{off}, 1}$, but in the $k_{\text{off}}$-case only for a time $1/(2k_{\text{off}, 1})$. This means that in the $k_{\text{off}}$-case, the edge rapidly converges more rapidly back to its equilibrium value.

In order to account for the strong anisotropy of myosin we observe, with myosin on AP-edges barely above cytosolic levels, purely by a dependence of $k_{\text{off}}$ on the junction orientation $\theta$, $k_{\text{off}}$ would have to be very large on junctions disaligned with the DV axis. This means that rotating junctions would rapidly lose their myosin as explained above, leading to no significant shift in the width and mean of the myosin angular distribution. Simulations similar to those shown in *Appendix 1—figure 3* confirm this argument.

## Principal axis of embryo-scale tension and turgor pressure

As mentioned in the 'Discussion' section, one possible candidate for the statically oriented myosin source is epithelial tension that myosin dynamics is known to be sensitive to *Fernandez-Gonzalez et al., 2009*. The direction of tension agrees with the direction of the inferred myosin source. Indeed, epithelial tension in the germband is known to be strongly anisotropic from laser ablation experiments, with higher tension on junctions parallel to the DV axis (*Munjal et al., 2015*). *Noll et al., 2020*, using an image-based force inference algorithm, confirmed that on the scale of the entire embryo, the epithelial tension aligns with the geometric DV axis, even after the onset of tissue flow. *Noll et al., 2020* also found that the distribution of junctional myosin closely matched the epithelial tension. Strikingly, most junctional myosin is balanced: ~80% of junctional myosin is involved in static force balance, i.e., it creates a net-zero local force.

Previous work cell-scale literature, e.g., *Munjal et al., 2015*, presented the anisotropic tension as a consequence of the anisotropy of junctional myosin. However, to set a static myosin orientation, the epithelial tension cannot be a pure readout of the current myosin distribution. One potential static contribution to tension anisotropy is the turgor pressure difference between the yolk within the blastoderm and the perivitlline space outside of it (e.g., visible during dorsal closure; *Lu et al., 2016*). This normal pressure is balanced by epithelial surface stress, much like the excess pressure in an inflated balloon. Pressure, stress, and geometry are linked by the Young–Laplace law. Due to the embryo's cylinder-like geometry, the resulting surface stress is anisotropic: in a pressurized cylinder with closed ends, the stress along the azimuthal axis is twice the stress along the height axis of the cylinder (*Audoly and Pomeau, 2010*). Fairly generally, excess internal pressure leads to anisotropic stress parallel to principal axes of curvature (*Audoly and Pomeau, 2010*; *Deserno, 2015*). Interestingly, blastodermic turgor pressure leading to cortical tension has already been shown to play a crucial role in mouse blastocyst development (*Chan et al., 2019*)

Finally, theoretical work, *Noll et al., 2020* (recently experimentally validated in *Gustafson et al., 2021*) has shown how epithelial tissue can support static tension even during viscous flow. Strain-rate based recruitment can drive junctional myosin to a balanced state (with zero net local forces), such as that required to balance turgor pressure.

## Additional data on mutants

*Appendix 1—figure 17* shows a kymograph of a contracting junction in WT and *eve*, illustrating that in *eve* mutants, myosin is still associated with junction contraction. *Appendix 1—figure 18* shows that the myosin distribution in *eve* is significantly less anisotropic than in WT, even if the anisotropy remains clear. To test that this difference is not due to better visibility of edges in $eve^{R13}$ due to myosin being visualized using Myosin::mCherry instead of a Myosin::GFP, we verified that this difference between WT and *eve* persists if fewer and fewer *eve* junctions are included (filtering by myosin intensity, excluding up to 3/4 detected junctions).

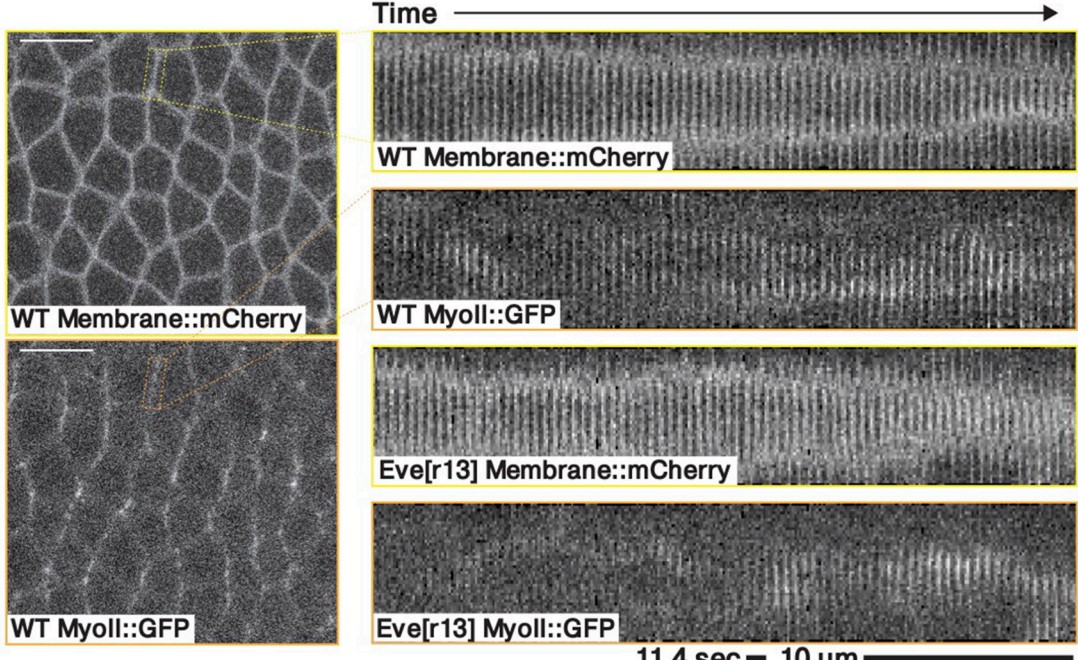

**Appendix 1—figure 17.** Kymograph of a contracting junction in a representative WT and a representative *eve* embryo. Both kymographs shows a junction in the germband ~10–20 minutes post VF initiations, marked with both a membrane and a myosin fluorescent tag.

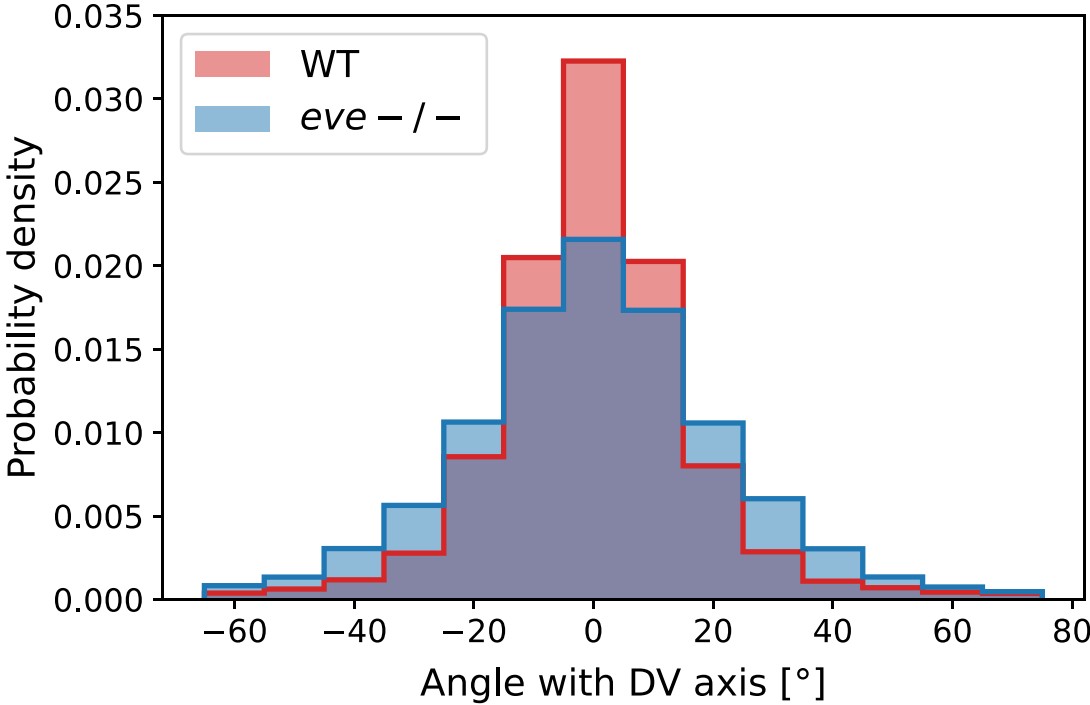

**Appendix 1—figure 18.** Histogram of myosin orientations in the germband of $N = 2$ Myosin::GFP WT $N = 2$ Myosin:mCherry eve embryos. Data corresponds to 15 min post VF initiations. For each embryo, more than 7000 edges are detected. The two-sided KS statistic (maximal area difference between cumulative distribution functions) is 0.076.

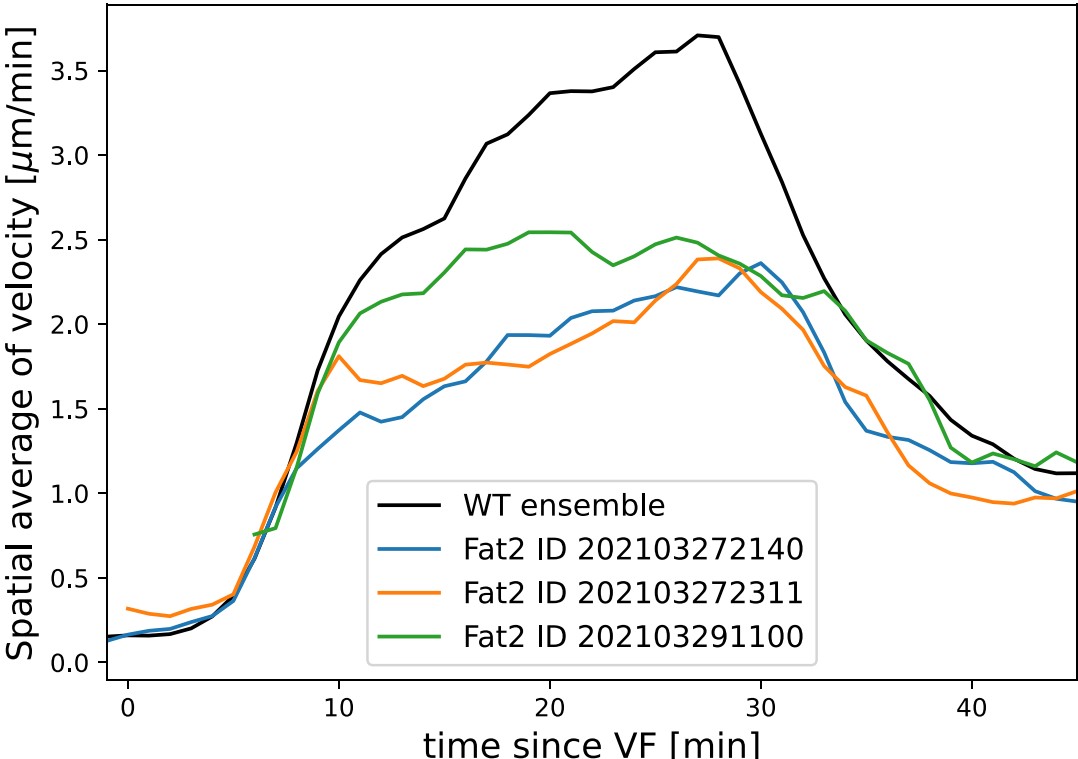

**Appendix 1—figure 19.** Average tissue flow velocity in WT and three *Fat2-RNAi* embryos. Flow in round embryos is noticeably reduced. All measurements computed using the induced metric to correct for any distortions of the cylindrical projections.

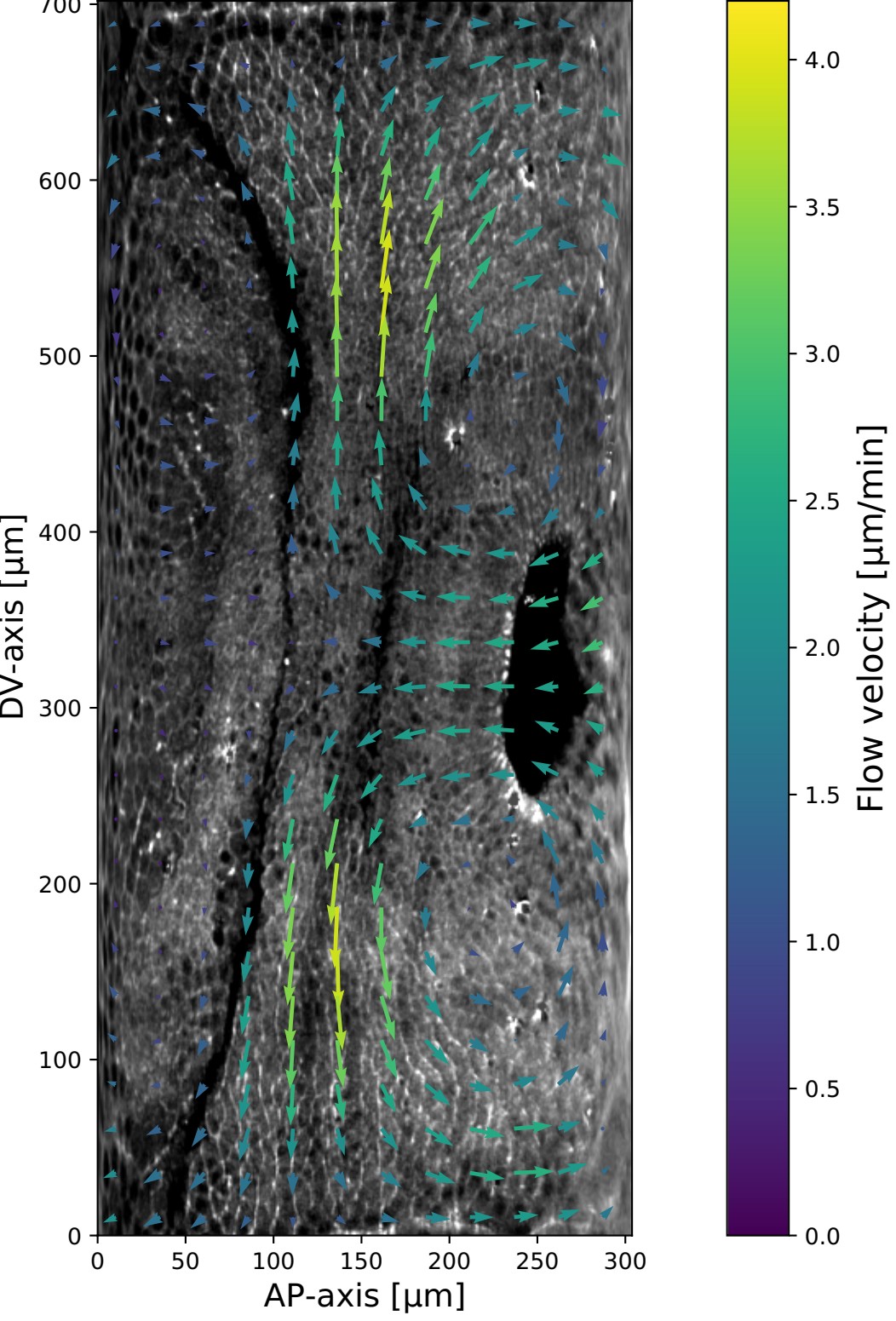

**Appendix 1—figure 20.** Myosin and particle image velocimetry (PIV) field on a representative *Fat2-RNAi* embryo. Time 10 min post VF initiation. The signal shown is the raw myosin signal, not subjected to the cytosolic normalization procedure, to show the embryo anatomy. The PIV vortices (regions of maximal vorticity) are removed from the regions with significant junctional myosin accumulation.

