## [Editor Report]

This article reports fundamental findings regarding spatiotemporal control of myosin-based force generation during *Drosophila* germband extension and is of considerable interest to our understanding of tissue morphogenesis during early development. Using quantitative imaging, mathematical modeling, and mutant analysis, the authors provide compelling evidence that myosin polarity patterns are not governed by pair-rule gene expression, but that a geometric cue promotes myosin II accumulation of vertically oriented junctions. The results challenge current views of how gene expression patterns control myosin II anisotropies and provide new testable hypotheses on the role and importance of tissue geometry.

---

## [Decision Letter]

**Decision letter after peer review:**

Thank you for submitting your article "Geometric control of Myosin-II orientation during axis elongation" for consideration by *eLife*. Your article has been reviewed by 3 peer reviewers, and the evaluation has been overseen by a Reviewing Editor and Anna Akhmanova as the Senior Editor. The following individual involved in the review of your submission has agreed to reveal their identity: Ed Munro (Reviewer #3).

The reviewers have discussed their reviews with one another, and the Reviewing Editor has drafted this to help you prepare a revised submission. All referees agree that this is a strong study reporting important findings regarding the dynamics of a developing *Drosophila* embryo, supported by rigorous quantitative analysis and modeling. There is a consensus amongst the referee to ask for additional data regarding the myosin lifetime in mutants, which would strongly support your predictions. The referees also made a number of comments and suggestions that will help improve the paper.

Essential revisions:

1) Explain more clearly in the main text how the data are used to compute the myosin lifetime (Reviewer 1, 1) and Reviewer 3, 1.b))

2) Report data on myosin lifetime for mutant (in particular Fat2-RNAi) embryos (Reviewer 1, 2) and Reviewer 3, 5))

3) Explain in the main text why more complex models can be ruled out (Reviewer 1, 5) and Reviewer 3, 2)).

4) Reformulate or temper some claims, in particular regarding changing change "PRGs" to "Runt" and "TLRs" to "Tartan" throughout the paper (Reviewer 1, 3)) and Lagrangian dynamics into something like "passive advection" (Reviewer 2).

5) Consider the possibility of using a symbolic regression technique to disprove a relationship more complex than the linear one you tested between PRG and myosin (reviewer 2).

6) Check for myosin enrichment in junctions that become aligned with the DV axis (Reviewer 3, 4).

Please also consider the other comments made by the reviewers, below.

*Reviewer #1 (Recommendations for the authors):*

1. Using mathematical modeling, the authors propose a myosin lifetime of 5 minutes. The experimentally measured myosin turnover rate in this study is just over 3 minutes. The authors should provide more information about the myosin lifetime parameter and comment on possible reasons for the approximately 1.5-fold difference between the myosin lifetime predicted by modeling and the time to full recovery measured using FRAP.

2. The authors should clarify the predictions of their model for eve mutant, twist mutant, and Fat2-RNAi embryos and perform FRAP experiments to experimentally test these predictions. The authors should apply their model to Fat2-RNAi, which provides the most direct test of their model that embryo geometry provides the static source that aligns myosin anisotropy with the DV axis. At face value, it is not clear how the results in Figure 4, which show largely normal myosin dynamics in a substantially geometrically altered embryo, argue for an instructive role of tissue geometry. The authors should include a more complete analysis of Fat2-RNAi embryos, including applying their model to Fat2-RNAi, analyzing myosin dynamics in the ventrolateral region of Fat2-RNAi embryos by FRAP, and quantifying the change in embryo DV width to document the effects on embryo geometry.

3. The authors extrapolate from their data on Runt to all pair-rule genes (PRGs) and from their data on Tartan to Toll-like receptors (TLRs). These assumptions are not well supported, as it cannot be assumed that all PRGs behave similarly, and Tartan is not only not a TLR, but it also has a different phenotype and function that (unlike TLRs) is restricted to compartment boundaries. As a result, there is no reason to assume that conclusions about Tartan will pertain to TLRs. The authors should change "PRGs" to "Runt" and "TLRs" to "Tartan" throughout the paper.

4. Claims that the myosin pattern is static, made in the Abstract and in parts of the main text, are not fully consistent with the authors' data. Figure 3b shows that the myosin pattern significantly shifts before realigning with the DV axis, and Figure 2k' shows that myosin is aligned with Runt stripe 6 until t=20 minutes, even though the Runt stripe shifts substantially relative to the DV axis. Myosin anisotropy that is out of alignment with the DV axis is also apparent from the images of posterior regions in Figure 2i' and 2j.

5. The last sentence of the Results states that "these results suggest that instead of instructing anisotropic myosin recruitment, PRGs influence the myosin anisotropy by regulating retention of myosin to junctions" is not well supported by the authors' data. Numerous studies, including by the authors, show that eve mutants have less cortical myosin and that myosin turnover is inversely correlated with myosin levels at the cortex. The authors' data are equally consistent with the alternative interpretation that PRGs instruct anisotropic myosin recruitment, with myosin dynamics later modulated by a myosin positive feedback mechanism. The authors should explicitly acknowledge alternative interpretations that are consistent with their data.

6. In the Materials and methods, Fly stocks, and genetics section, "hemizygous control embryos lacked halo" should be changed to "heterozygous control embryos did not show the halo phenotype". The authors should describe the Fat2 RNAi method used. References describing the generation of all antibodies should be cited, and the peptide sequence used to make the Tartan antibody should be described.

7. The location of dorsal and ventral regions should be indicated in all figures, as it is not clear if all embryos are similarly oriented. In Figure 2f and 2g, the correlation coefficients are labeled with the same value. In Figure 2a, measured is misspelled.

*Reviewer #2 (Recommendations for the authors):*

Overall, I think this is a very good paper, and subject to a couple of revisions would recommend it for publication.

Suggestions:

As I understand it, your experimental measurements include both the full velocity field and the various director fields. Given that a big element of this paper is demonstrating the validity (or lack thereof) of a Lagrangian description of the dynamics, I think it would strengthen the paper to include some direct analysis of the upper convected derivative of both the PRGs and myosin director fields. For instance, a comparison of the magnitude of this quantity for PRGs and myosin at t=7 and t=22 would be helpful in understanding the differences in dynamics between the two.

Since this is a case of tensor transport over a curved surface, there should be some discussion of the possible role of parallel transport. The region of interest in the WT *Drosophila* embryo is essentially a cylinder you can probably rule out via a quick back-of-the-envelope calculation, but I imagine that there might be a non-negligible effect on the Fat2 mutant, subject to what the flow lines look like in the high vorticity regions.

The language regarding Lagrangian dynamics should be tightened up a bit. Formally, Lagrangian should be used in reference to the Lagrangian frame of reference or coordinate system of the fluid. In my experience, the phrase "Lagrangian pattern," as in the last sentence of the first paragraph, is not really well defined and is being used here to describe what should just be called passive advection. Likewise, it does not make much sense to say that PRGs and TLRs are in a flowing frame of reference, as stated in the last paragraph of the introduction. Any phenomena can be in any frame of reference you want – it would be more correct to say PRGs and TLRs are easy to describe in that particular frame. At these points and in a couple of other places where Lagrangian flows are discussed, the language should be made more precise.

The analysis relating to Equation 1 is fine in itself, but the conclusions drawn from it are not especially convincing, as it is easy to imagine that a nonlinear relationship exists between myosin and PRGs. I think it would be easy to improve this result by applying a good symbolic regression technique to your existing data to see if that likewise fails to generate a clear function for myosin.

In section SE, m is used for two different quantities: to symbolize the local myosin alignment tensor in (13) but also for the angular distribution throughout subsection SE1. Maybe substitute the first m with Q or another symbol.

It is a slight overclaim to say you can independently modulate model parameters in vivo. Myosin kinetics, geometry, and vorticity are not independent, and none of your experiments actually let you alter one while fixing all the rest.

Sometimes the authors write "germband" and other times "germ band" – there is not really a correct spelling but one should be selected and used consistently in the paper.

Figure 3h is a little confusing at first glance. We are meant to compare h, h', and h', but the bar colors mean different things across the panels and there are vortex streamlines in the back of h' even though some of the data on that plot represents irrotational flow.

*Reviewer #3 (Recommendations for the authors):*

(1) The comparisons of model predictions and experimental observations are presented in a way that I found (and I expect other readers will find) confusing. It would be useful to clarify in the main text (with links to SI as appropriate):

(a) What data are the authors fitting to obtain estimates of myosin lifetime?

I could not find any description of this, either in the main text or the SI. This would be especially important for the analysis of eve mutant embryos where they are drawing a strong conclusion that the myosin lifetime is reduced in these mutants.

(b) What additional predictions of the model are they then comparing to additional data?

For example, the direct comparison of observed and simulated distributions of MRJ orientations shown in Figure 3h is very clear and compelling. It would be helpful to show similar comparisons for Figures 3b and 3i to back up what currently seems like vague and qualitative statements in the main text.

(2) Several questions arose as I was reading the main text. I later discovered that these are addressed in the SI. Because I suspect other readers will have similar questions – I suggest the authors address these questions briefly in the main text and refer to the more detailed discussion given in the SI. These include:

(a) To what extent do local reorientation of MRJ's, e.g due to anisotropic junction shortening, T1 transitions, cell divisions, etc affect the orientation distribution of MRJs during GBE?

(b) In principle, a model based on modulation of myosin detachment by a static geometric cue could also explain the steady state alignment of MRJ's with the DV axis. Can the author's observations exclude this alternative possibility?

(3) The data in Figure 4 showing the analysis of different mutants is also presented in a somewhat confusing way. The opening sentences of the section make it sound like the authors are using existing knowledge of certain mutants to systematically and independently manipulate specific parameters in the model. But it seems like what they are really showing (for twist and eve) is that mutant phenotypes could be explained as tuning variants of the simple model (with different vorticities and myosin detachment times).

By contrast, for fat-2, they are not testing the static source model at all – instead, they are extending an earlier conclusion about the insufficiency of PR and TLR expression patterns that do explain myosin II anisotropies.

(4) A key prediction of the static source model is that myosin II would begin to accumulate on junctions that rotate into alignment with the DV axis in regions of high vorticity (i.e. stripe 6). Would it be possible to test this with their existing data by tracking (perhaps even by hand) a subset of junctions that become so aligned?

(5) The model fits to eve mutant data suggest that there is a reduction in the myosin II detachment rate. It would be awesome, if feasible, to test this directly with FRAP experiments.

---

## [Author Response]

Reviewer #1 (Recommendations for the authors):1. Using mathematical modeling, the authors propose a myosin lifetime of 5 minutes. The experimentally measured myosin turnover rate in this study is just over 3 minutes. The authors should provide more information about the myosin lifetime parameter and comment on possible reasons for the approximately 1.5-fold difference between the myosin lifetime predicted by modeling and the time to full recovery measured using FRAP.

We have added additional discussion of the myosin lifetime parameter τ in the main text and SI:

“This value [of τ = 5 min,] is qualitatively similar, but somewhat larger, than the FRAP-measured myosin lifetime. As discussed in the SI, this is because FRAP measures the time individual motors remain on a junction, while τ measures how long the total myosin level on a junction persists. The latter time can be larger if factors which affect myosin levels, such as kinases, are longer-lived than individual motors.”

2. The authors should clarify the predictions of their model for eve mutant, twist mutant, and Fat2-RNAi embryos and perform FRAP experiments to experimentally test these predictions. The authors should apply their model to Fat2-RNAi, which provides the most direct test of their model that embryo geometry provides the static source that aligns myosin anisotropy with the DV axis. At face value, it is not clear how the results in Figure 4, which show largely normal myosin dynamics in a substantially geometrically altered embryo, argue for an instructive role of tissue geometry. The authors should include a more complete analysis of Fat2-RNAi embryos, including applying their model to Fat2-RNAi, analyzing myosin dynamics in the ventrolateral region of Fat2-RNAi embryos by FRAP, and quantifying the change in embryo DV width to document the effects on embryo geometry.

We have rewritten the section regarding mutant embryos for clarification. As another reviewer suggested, the main point of the *eve* and *twist* mutants is that their myosin dynamics can be well explained by tuning the quantities within out mathematical model (myosin lifetime for *eve*, vorticity for *twist*), both of which we measure (myosin lifetime by FRAP, vorticity by PIV)

“Harnessing the genetic toolkit available in the *Drosophila* model system, we show that we can account for the behavior of mutants by modulating parameters of our mathematical model – vorticity, myosin kinetics, and geometry (Figures 4a-a'').”

Next, we explain more clearly what we use the *Fat2-RNAi* data for:

“We originally planned to use *Fat2-RNAi* to modify the direction of the proposed static source, the geometric DV-axis. However, since *Fat2-RNAi* eggs remain highly rationally symmetric about the AP axis, the direction of the geometric DV axis is not significantly affected. Instead, this mutant provided an opportunity to test if PRG stripes would change in the same way as MRJs around the ectopically extended DV circumference”

The static source in our model is geometrically defined as the axis of maximum curvature. Due to the strong rotational symmetry of *Fat2-RNAi* eggs, this axis always points along the azimuthal axis with very weak deviation (1-2 degrees), like in WT. We quantifiy the change in DV width:

“[W]e created nearly spherical embryos with a shorter, but variable length, and up to 30\% extended DV circumference”

Finally, we recorded new data, carrying out FRAP experiments in *Fat2-RNAi* embryos. Their lifetime is not significantly different from WT, as expected from their intact AP patterning system [Chanet 2017].

3. The authors extrapolate from their data on Runt to all pair-rule genes (PRGs) and from their data on Tartan to Toll-like receptors (TLRs). These assumptions are not well supported, as it cannot be assumed that all PRGs behave similarly, and Tartan is not only not a TLR, but it also has a different phenotype and function that (unlike TLRs) is restricted to compartment boundaries. As a result, there is no reason to assume that conclusions about Tartan will pertain to TLRs. The authors should change "PRGs" to "Runt" and "TLRs" to "Tartan" throughout the paper.

We have edited the manuscript as suggested by reviewer 1, changing “TLRs” to “Tartan”, given that our experimental results mainly pertain to Tartan, for which the best antibody was available. Further, we have added SI Figure 10, which uses co-stains of the PRGs Even-Skipped, Fushi-Taratzu, Hairy, Paired, and Sloppy-Paired on the one, and Runt on the other hand. These co-stains show that during GBE, the relative position of all PRGs remains parallel to the Runt stripes, and therefore show that all PRGs imaged are advected by the flow in the same way as Runt. We have added the following discussion in the main text to highlight this issue:

”Since this [striped ]pattern of expression is stereotypic for the PRGs known to have the largest individual effects on GBE,, we adopted Runt as our representative PRG. In SI Figure 10 and in Figure 1d, we show that during GBE, the stripes of the PRGS Runt, Eve, Ftz, Paired, Sloppy-Paired, and Hairy remain parallel throughout GBE flow, and are transported by tissue flow in the same way. Therefore, it is sufficient to study the advection behavior of only one of them. Similarly, as a proxy for the TLRs, we chose Tartan, a leucine-rich-repeat receptor downstream of the PRGs which has been implicated in directing myosin anisotropy in concert with the TLRs 2, 6, and 8, due to the availability of a high-quality antibody.”

We hope this addresses the reviewer’s concern.

4. Claims that the myosin pattern is static, made in the Abstract and in parts of the main text, are not fully consistent with the authors' data. Figure 3b shows that the myosin pattern significantly shifts before realigning with the DV axis, and Figure 2k' shows that myosin is aligned with Runt stripe 6 until t=20 minutes, even though the Runt stripe shifts substantially relative to the DV axis. Myosin anisotropy that is out of alignment with the DV axis is also apparent from the images of posterior regions in Figure 2i' and 2j.

We have edited the language in the abstract and the Discussion section to reflect that the myosin pattern is not completely static. Indeed, one of the key points of the paper is the explanation of the dynamics of deviation of the myosin orientation from the DV axis. The abstract now reads:

“In contrast, myosin anisotropy orientation remained approximately static, and was only weakly deflected from the stationary dorsal-ventral axis of the embryo.”

5. The last sentence of the Results states that "these results suggest that instead of instructing anisotropic myosin recruitment, PRGs influence the myosin anisotropy by regulating retention of myosin to junctions" is not well supported by the authors' data. Numerous studies, including by the authors, show that eve mutants have less cortical myosin and that myosin turnover is inversely correlated with myosin levels at the cortex. The authors' data are equally consistent with the alternative interpretation that PRGs instruct anisotropic myosin recruitment, with myosin dynamics later modulated by a myosin positive feedback mechanism. The authors should explicitly acknowledge alternative interpretations that are consistent with their data.

We now acknowledge alternative interpretations of the data at the end of the Results section:

“One possibility is that PRGs play a role in directing myosin anisotropy in an initial phase, with control over myosin orientation transferred to a static cue once flow starts. However, this hypothesis is not supported by the lack of linear correlation between PRGs and the initial myosin pattern, the results in *Fat2-RNAi* embryos, and the results of [Gustafson 2021], which found that mechanical cues can explain the early myosin pattern.”

6. In the Materials and methods, Fly stocks, and genetics section, "hemizygous control embryos lacked halo" should be changed to "heterozygous control embryos did not show the halo phenotype". The authors should describe the Fat2 RNAi method used. References describing the generation of all antibodies should be cited, and the peptide sequence used to make the Tartan antibody should be described.

The Materials and methods section was edited as requested. We cite references for the Fat2-RNAi method, where we followed the work of Chanet et al. 2017, and indicate the sources for all the antibodies used.

7. The location of dorsal and ventral regions should be indicated in all figures, as it is not clear if all embryos are similarly oriented. In Figure 2f and 2g, the correlation coefficients are labeled with the same value. In Figure 2a, measured is misspelled.

We fixed the erroneous correlation coefficient in Figure 2f. We have clarified in the text that whole-embryo images (3d and 2d) are always oriented as indicated in Figure 1b, and added clarifications about the orientation of images showing sub-regions in the figure legends where not already present.

Reviewer #2 (Recommendations for the authors):Overall, I think this is a very good paper, and subject to a couple of revisions would recommend it for publication.Suggestions:As I understand it, your experimental measurements include both the full velocity field and the various director fields. Given that a big element of this paper is demonstrating the validity (or lack thereof) of a Lagrangian description of the dynamics, I think it would strengthen the paper to include some direct analysis of the upper convected derivative of both the PRGs and myosin director fields. For instance, a comparison of the magnitude of this quantity for PRGs and myosin at t=7 and t=22 would be helpful in understanding the differences in dynamics between the two.

We have added an analysis of the convective derivative of the myosin and Runt pattern in SI Figure 11 (SI Sect. E), and reference this analysis in the main text:

“Finally, in SI Figure 11 we show that in accordance with the above analyses [Figure 2], the rate of change of the Runt pattern is much lower in the Lagragian frame of reference which flows with the tissue than in a static frame of reference, whereas the opposite is true for the myosin orientation.”

Since this is a case of tensor transport over a curved surface, there should be some discussion of the possible role of parallel transport. The region of interest in the WT *Drosophila* embryo is essentially a cylinder you can probably rule out via a quick back-of-the-envelope calculation, but I imagine that there might be a non-negligible effect on the Fat2 mutant, subject to what the flow lines look like in the high vorticity regions.

We have added a significant section discussing the role of curvature and differential geometry in the calculation of the transport of the myosin tensor. As this material is relatively technical, we have included it in the SI as Sect. G2b. In brief, curvature does not play a significant role even in Fat2 embryos, since the differential geometric notion that best captures advection is Lie transport, not parallel transport, which is not directly dependent on the metric of the curved embryo surface. Further, we note that in the Fat2 embryos, the regions of high vorticity and high myosin anisotropy overlap very little (SI Figure 16), so that the influence of vorticity here is in any case limited.

The language regarding Lagrangian dynamics should be tightened up a bit. Formally, Lagrangian should be used in reference to the Lagrangian frame of reference or coordinate system of the fluid. In my experience, the phrase "Lagrangian pattern," as in the last sentence of the first paragraph, is not really well defined and is being used here to describe what should just be called passive advection. Likewise, it does not make much sense to say that PRGs and TLRs are in a flowing frame of reference, as stated in the last paragraph of the introduction. Any phenomena can be in any frame of reference you want – it would be more correct to say PRGs and TLRs are easy to describe in that particular frame. At these points and in a couple of other places where Lagrangian flows are discussed, the language should be made more precise.

We have followed the suggestions of the reviewer in regards to the language used and replaced “Lagrangian pattern” etc by “passively advected pattern” or similar formulation. We now use the term “Lagrangian” only when talking about reference frames.

The analysis relating to Equation 1 is fine in itself, but the conclusions drawn from it are not especially convincing, as it is easy to imagine that a nonlinear relationship exists between myosin and PRGs. I think it would be easy to improve this result by applying a good symbolic regression technique to your existing data to see if that likewise fails to generate a clear function for myosin.

We have added an extensive analysis of more general, non-linear models for a potential relationship between PRG patterns and myosin anisotropy, in SI Sect. F1. We have attempted to use symbolic regression, as implemented in python by the package gplearn, but found it difficult to get to work, even on test examples. Instead, we considered a class of regression models with polynomial features, where the relationship between PRGs and myosin can be non-linear. As explained in SI Sect. E1, we find that they too fail to well explain the observed myosin pattern.

In section SE, m is used for two different quantities: to symbolize the local myosin alignment tensor in (13) but also for the angular distribution throughout subsection SE1. Maybe substitute the first m with Q or another symbol.

As suggested, we have changed notation to remove ambiguity – the myosin tensor and the myosin angular distribution are now represented by different symbols.

It is a slight overclaim to say you can independently modulate model parameters in vivo. Myosin kinetics, geometry, and vorticity are not independent, and none of your experiments actually let you alter one while fixing all the rest.

We have edited the language in the section discussing mutants:

“Harnessing the genetic toolkit available in the *Drosophila* model system, we show that we can account for the behavior of mutants by modulating parameters of our mathematical model – vorticity, myosin kinetics, and geometry (Figures 4a-a'').”

Sometimes the authors write "germband" and other times "germ band" – there is not really a correct spelling but one should be selected and used consistently in the paper.

We have adopted a consistent spelling for “germband”.

Figure 3h is a little confusing at first glance. We are meant to compare h, h', and h', but the bar colors mean different things across the panels and there are vortex streamlines in the back of h' even though some of the data on that plot represents irrotational flow.

We have changed the optics of Figure 3 to improve clarity. The histogram color in h-h’’ and i now always matches the vorticity, i.e. red for no vorticity, blue for high vorticity.

Reviewer #3 (Recommendations for the authors):(1) The comparisons of model predictions and experimental observations are presented in a way that I found (and I expect other readers will find) confusing. It would be useful to clarify in the main text (with links to SI as appropriate):(a) What data are the authors fitting to obtain estimates of myosin lifetime?I could not find any description of this, either in the main text or the SI. This would be especially important for the analysis of eve mutant embryos where they are drawing a strong conclusion that the myosin lifetime is reduced in these mutants.(b) What additional predictions of the model are they then comparing to additional data?

a) We have included the following description of the fitting procedure:

“First, we computed the angle between the orientation of myosin anisotropy and the DV axis, adjusted for the rate of tissue rotation by solving Equation 4 (see SI for mathematical definition). Using an ensemble of N=5 embryos, we then fitted the parameter τ by minimizing the average difference between the myosin orientation predicted from Equation 4 and the observed orientation across the germband during convergent extension”

The same fitting protocol was used for the Eve mutant analysis.

b) We have changed Figure 3 to make the comparison between data and model prediction clearer. We have edited panel 3b) to include a direct prediction of the spatial average myosin/DV axis angle. For panel 3i, the histograms quantifying the recovery of myosin/DV alignment after vorticity turns off, the theoretical histograms of 3h’’ are the relevant theory comparison point. We have edited the layout and design of the figure to make this visually clear, and added an explanation in the main text.

For example, the direct comparison of observed and simulated distributions of MRJ orientations shown in Figure 3h is very clear and compelling. It would be helpful to show similar comparisons for Figures 3b and 3i to back up what currently seems like vague and qualitative statements in the main text.(2) Several questions arose as I was reading the main text. I later discovered that these are addressed in the SI. Because I suspect other readers will have similar questions – I suggest the authors address these questions briefly in the main text and refer to the more detailed discussion given in the SI. These include:(a) To what extent do local reorientation of MRJ's, e.g due to anisotropic junction shortening, T1 transitions, cell divisions, etc affect the orientation distribution of MRJs during GBE?(b) In principle, a model based on modulation of myosin detachment by a static geometric cue could also explain the steady state alignment of MRJ's with the DV axis. Can the author's observations exclude this alternative possibility?

As suggested, we now address both questions briefly in the main text, with a reference to the SI for more details:

a) “In addition to tissue rotation, junctions could in principle be rotated by tissue strain. This is however not the case in the germ band, because the majority of tissue strain is due to cell rearrangement. Junction shortening or lengthening also does not affect junction orientation. For a detailed discussion, see SI Sect. F”

b) “It is also possible to consider a model where the myosin recruitment rate is constant and instead, the detachment rate is modulated according to junction orientation. We find that such a model cannot account for the observed deflection of myosin orientation (SI Sect.G.2d)”

(3) The data in Figure 4 showing the analysis of different mutants is also presented in a somewhat confusing way. The opening sentences of the section make it sound like the authors are using existing knowledge of certain mutants to systematically and independently manipulate specific parameters in the model. But it seems like what they are really showing (for twist and eve) is that mutant phenotypes could be explained as tuning variants of the simple model (with different vorticities and myosin detachment times).

We have changed the text of the manuscript to clarify:

“Harnessing the genetic toolkit available in the *Drosophila* model system, we show that we can account for the behavior of mutants by modulating parameters of our mathematical model – vorticity, myosin kinetics, and geometry (Figures 4a-a'').”

“We originally planned to use *Fat2-RNAi* to modify the direction of the proposed static source, the geometric DV-axis. However, since *Fat2-RNAi* eggs remain highly rationally symmetric about the AP axis, the direction of the geometric DV axis is not significantly affected. Instead, this mutant provided an opportunity to test if PRG stripes would change in the same way as MRJs around the ectopically extended DV circumference.”

By contrast, for fat-2, they are not testing the static source model at all – instead, they are extending an earlier conclusion about the insufficiency of PR and TLR expression patterns that do explain myosin II anisotropies.(4) A key prediction of the static source model is that myosin II would begin to accumulate on junctions that rotate into alignment with the DV axis in regions of high vorticity (i.e. stripe 6). Would it be possible to test this with their existing data by tracking (perhaps even by hand) a subset of junctions that become so aligned?

The referee is correct in pointing out a clear prediction of our static-source model. Unfortunately, our current datasets are not suitable for tracking because of they have a 1-minute time resolution and only show myosin, making myosin-poor junctions hard to see. However, previous work by the Zallen lab [Farrell et al. 2017, Figure 6] actually carries out the analysis proposed by the referee with more suitable confocal data showing both myosin and Par-3 (a junctional protein enriched on horizontal, myosin-poor junctions). The authors find, exactly as expected, that there are junctions which become enriched in myosin once they rotate into alignment with the DV axis. The local observations of Farrell and colleagues complement the comprehensively global and quantitative analysis presented in our work. We have edited the manuscript to draw the reader’s attention towards the work of Farrell et al.:

“Our global, comprehensive analysis confirms and extends previous local, tracking-based observations which showed that junctions can remodel their myosin levels as they rotate [Farrell et al. 2017]”

(5) The model fits to eve mutant data suggest that there is a reduction in the myosin II detachment rate. It would be awesome, if feasible, to test this directly with FRAP experiments.

To clarify, in our initial submission, we hypothesized an *increased* detachment rate (i.e. a decreased lifetime) of myosin II on junctions in *eve* mutants. We have now carried out FRAP experiments in Eve mutants. The results appear to be compatible with our initial hypothesis. However, the overall dynamics of the FRAP curve is altered compared to the WT (with a single-exponential recovery, as compared to the biphasic behavior observed in WT), making the interpretation somewhat ambiguous. This is discussed in detail in a new SI section about FRAP mutant data, SI Sect. D. For completeness, we have also included new FRAP experiments in *Fat2-RNAi* embryos, although we do not refer to this data in the main text.

References:

D.L Farrell, O. Weitz, M. O. Magnasco, and J. A. Zallen. *SEGGA: a toolset for rapid automated analysis of epithelial cell polarity and dynamics*, Development 144, 1725 (2017).